# Learning Distributed and Fair Policies for Network Load Balancing as Markov Potential Game

**Zhiyuan Yao**[*]
École Polytechnique, Cisco Systems
`zhiyuan.yao@polytechnique.edu`

**Zihan Ding**[*]
Princeton University
`zihand@princeton.edu`

## Abstract

This paper investigates the network load balancing problem in data centers (DCs) where multiple load balancers (LBs) are deployed, using the multi-agent reinforcement learning (MARL) framework. The challenges of this problem consist of the heterogeneous processing architecture and dynamic environments, as well as limited and partial observability of each LB agent in distributed networking systems, which can largely degrade the performance of in-production load balancing algorithms in real-world setups. Centralised-training-decentralised-execution (CTDE) RL scheme has been proposed to improve MARL performance, yet it incurs – especially in distributed networking systems, which prefer distributed and plug-and-play design schemes – additional communication and management overhead among agents. We formulate the multi-agent load balancing problem as a Markov potential game, with a carefully and properly designed workload distribution fairness as the potential function. A fully distributed MARL algorithm is proposed to approximate the Nash equilibrium of the game. Experimental evaluations involve both an event-driven simulator and real-world system, where the proposed MARL load balancing algorithm shows close-to-optimal performance in simulations, and superior results over in-production LBs in the real-world system.

## 1 Introduction

In cloud data centers (DCs) and distributed networking systems, servers are deployed on infrastructures with multiple processors to provide scalable services [1]. To optimise workload distribution and reduce additional queuing delay, load balancers (LBs) play a significant role in such systems. State-of-the-art network LBs rely on heuristic mechanisms [2–5] under the low-latency and high-throughput constraints of the data plane. However, these heuristics are not adaptive to dynamic environments and require human interventions, which can lead to most painful mistakes in the cloud – mis-configurations. RL approaches have shown performance gains in distributed system and networking problems [6–9], yet applying RL on the network load balancing problem is challenging.

First, unlike traditional workload distribution or task scheduling problem [6, 7], network LBs have limited observations over the system, including task sizes and actual server load states. Being aware of only the number of tasks they have distributed, servers can be overloaded by collided elephant tasks and have degraded quality of service (QoS).

Second, to guarantee high service availability in the cloud, multiple LBs are deployed in DCs. Network traffic is split among all LBs. This multi-agent setup makes LBs have only partial observation over the system.

Third, modern DCs are based on heterogeneous hardware and elastic infrastructures [10], where server capacities vary. It is challenging to assign correct weights to servers according to their actual

---

[*]Equal contribution.

36th Conference on Neural Information Processing Systems (NeurIPS 2022).

processing capacities, and this process conventionally requires human intervention – which can lead to error-prone configurations [3, 5].

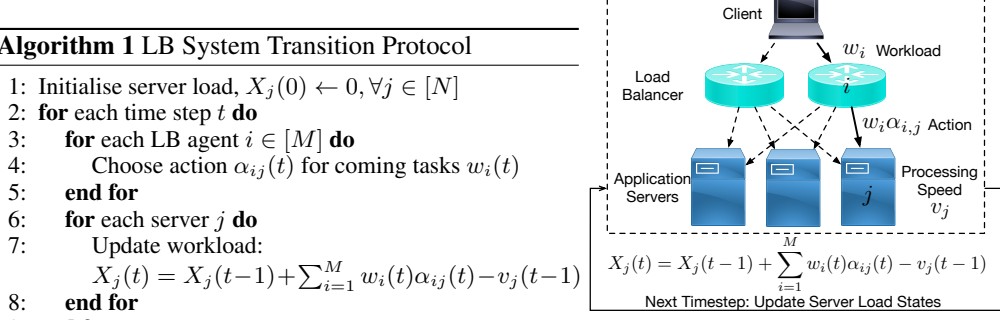

**Algorithm 1** LB System Transition Protocol

1: Initialise server load, $X_j(0) \leftarrow 0, \forall j \in [N]$
2: **for** each time step $t$ **do**
3:    **for** each LB agent $i \in [M]$ **do**
4:       Choose action $\alpha_{ij}(t)$ for coming tasks $w_i(t)$
5:    **end for**
6:    **for** each server $j$ **do**
7:       Update workload:
      $X_j(t) = X_j(t-1) + \sum_{i=1}^{M} w_i(t)\alpha_{ij}(t) - v_j(t-1)$
8:    **end for**
9: **end for**

Figure 1: Network load balancing.

Last but not least, given the low-latency and high-throughput constraints in the distributed networking setup, the interactive training procedure of RL models and the centralised-training-decentralised-execution (CTDE) scheme [11] can incur additional communication and management overhead.

In this paper, we study the network load balancing problem in multi-agent game theoretical approach, by formulating it as a Markov potential game through specifying the proper reward function, namely variance-based fairness. We propose a distributed Multi-Agent RL (MARL) network load balancing mechanism that is able to exploit asynchronous actions based only on local observations and inferences. Load balancing performance gains are evaluated based on both event-based simulations and real-world experiments[2].

## 2 Related Work

**Network Load Balancing Algorithms.** The main goal of network LBs is to *fairly* distribute workloads across servers. The system transition protocol of network load balancing system is described in Alg. 1 and depicted in Fig. 1. Existing load balancing algorithms are sensitive to partial observations and inaccurate server weights. Equal-Cost Multi-Path (ECMP) LBs randomly assign servers to new requests [12–14], which makes them agnostic to server load state differences. Weighted-Cost Multi-Path (WCMP) LBs assign weights to servers proportional to their provisioned resources (*e.g.* CPU power) [3, 15–17]. However, the statically assigned weights may not correspond to the actual server processing capacity. As depicted in Fig. 2a, servers with the same IO speed yet different CPU capacities have different actual processing speed when applications have different resource requirements. Active WCMP (AWCMP) is a variant of WCMP and it periodically probe server utilisation information (CPU/memory/IO usage) [5, 18]. However, active probing can cause delayed observations and incur additional control messages, which degrades the performance of distributed networking systems. Local Shortest Queue (LSQ) assigns new requests to the server with the minimal number of ongoing networking connections that are *locally* observed [19, 20]. It does not concern server processing capacity differences. Shortest Expected Delay (SED) derives the "expected delay" as locally observed server queue length divided by statically configured server processing speed [2]. However, LSQ and SED are sensitive to partial observations and misconfigurations. As depicted in Fig. 2b, the QoS performance of each load balancing algorithm degrades from the ideal setup (global observations and accurate server weight configurations) when network traffic is split across multiple LBs or server weights are mis-configured[3], which prevails in real-world cloud DCs.

In this paper, we propose a distributed MARL-based load balancing algorithm that considers dynamically changing queue lengths (*e.g.* sub-ms in modern DC networks [21]), and autonomously adapts to actual server processing capacities, with no additional communications among LB agents or servers.

**Markov Potential Games.** A potential game (PG) [22–25] has a special function called *potential function*, which specifies a property that any individual deviation of the action for one player will

---

[2]Source code and data of both simulation and real-world experiment are open-sourced at https://github.com/ZhiyuanYaoJ/MARLLB.

[3]The stochastic Markov model of the simulation is detailed in the App. A

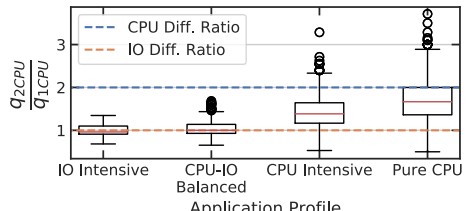
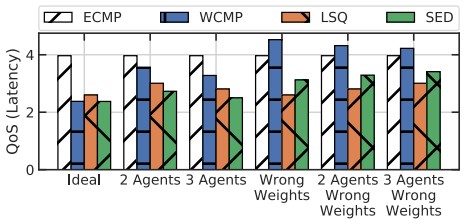

(a) It is hard to accurately estimate the actual server processing speeds since it depends on both provisioned resources, and application profiles (App. E.1).

(b) The performance of existing network load balancing algorithms degrades when observation becomes partial with multi-agents and weights are mis-configured.

Figure 2: Existing network load balancing algorithms are sub-optimal under real-world setups.

Table 1: Trade-offs among the probing frequency, measurement quality, and communication overhead.

| Probing Frequency (/s) | | 2.22 | 2.86 | 4.00 | 6.67 | 20.00 |
|---|---|---|---|---|---|---|
| RMSE | CPU (%) | 48.33 | 44.56 | 39.84 | 32.65 | 21.97 |
| | #Job | 2.07 | 1.85 | 1.61 | 1.31 | 0.91 |
| Spearman's Corr. | CPU (%) | 0.28 | 0.40 | 0.52 | 0.68 | 0.85 |
| | #Job | 0.47 | 0.56 | 0.66 | 0.77 | 0.89 |
| Communication Overhead (kbps) | 2LB-7server | 2.15 | 2.76 | 3.86 | 6.44 | 9.32 |
| | 6LB-20server | 18.40 | 23.66 | 33.12 | 55.20 | 165.60 |

change the value of its own and the potential function equivalently. A desirable property of PG is that pure NE always exists and coincides with the maximum of potential function in norm-form setting. Self-play [26] is provably converged for PG. Markov games (MG) is an extension of normal-form game to a multi-step sequential setting. A combination of PG and MG yields the Markov potential games (MPG) [27, 28], where pure NE is also proved to exist. Some algorithms [27, 29, 30] lying in the intersection of game theory and reinforcement learning are proposed for MPG. For example, independent nature policy gradient is proved to converge to Nash equilibrium (NE) for MPG [27].

**Multi-Agent RL**. MARL [31] has been viewed as an important avenue for solving different types of games in recent years. For cooperative settings, a line of work based on joint-value factorisation have been proposed, involving VDN [32], COMA [11], MADDPG [33], and QMIX [34]. For these works, a global reward is assigned to players within the team, but individual policies are optimised to execute individual actions, known as the CTDE setting. MPG satisfies the assumptions of the value decomposition approach, with the well-specified potential function as the joint rewards. However, deploying CTDE RL models in real-world distributed system incurs additional communication latency and management overhead for synchronising agents and aggregating trajectories. These additional management and communication overheads can incur substantial performance degradation – constrained throughput and increased latency – especially in data center networks. As listed in Table 1, when we use active probing to measure server utilisation information, higher probing frequencies give improved measurement quality–in terms of CPU usage and number of on-going jobs on the servers. However, higher probing frequencies also incur increased communication overhead, especially in large-scale data center networks. The detailed experimental setups, as well as both qualitative and quantitative analysis of the impact of communication overhead, are described in Sec. E.2.2. By leveraging the special structure of MPG, independent learning approach can be more efficient due to the decomposition of the joint state and action spaces, which is leveraged in the proposed methods. Methods like MATRPO [35], IPPO [36] follow a fully decentralised setting, but for general cooperative games.

In terms of the distribution fairness, FEN [37] is proposed as a decentralised approach for fair reward distribution in multi-agent systems. They defined the fairness as coefficient of variation and decompose it for each individual agent. Another work [38] proposes a decentralised learning method for fair policies in cooperative games. However, the decentralised learning manner in these methods are not well justified, while in this paper the load balancing problem is formally characterised as a MPG and the effectiveness of distributed training is verified.

## 3 Methods

### 3.1 Problem Description

We formulate the load balancing problem into a discrete-time dynamic game with strong distributed and concurrent settings, where no centralised control mechanism exists among agents. We let $M$ denote the number of LB agents ($[M]$ denotes the set of LB agents $\{1, \ldots, M\}$) and $N$ denote the number of servers ($[N]$ denotes the set of servers $\{1, \ldots, N\}$). At each time step (or round) $t \in H$ in a horizon $H$ of the game, each LB agent $i$ receives a workload $w_i(t) \in W$, where $W$ is the workload distribution, and the LB agent assigns a server to the task using its load balancing policy $\pi_i \in \Pi$, where $\Pi$ is the load balancing policy profile. At each time-step $t$, a LB agent $i$ takes an action $a_i(t) = \{a_{ij}(t)\}_{j=1}^N$, according to which the tasks $w_i(t)$ are assigned with distribution $\alpha_i(t)$. $\alpha_{ij}(t)$ is the probability mass of assigning tasks to server $j$, $\sum_{j=1}^N \alpha_{ij}(t) = 1$. Therefore, at each time step, the workload assigned to server $j$ by the $i$-th LB is $w_i(t)\alpha_{ij}(t)$. During each time interval, each server $j$ is capable of processing a certain amount of workload $v_j$ based on the property of each server (*e.g.* provisioned resources including CPU, memory, etc. ). We have server load state (remaining workload to process) $X_j(T) = \sum_{t=0}^T \max\{0, \sum_{i=1}^M w_i(t)\alpha_{ij}(t) - v_j\} = \max\{0, \sum_{t=0}^T \sum_{i=1}^M w_i(t)\alpha_{ij}(t) - v_j T\} = \sum_{i=1}^M X_{ij}(T)$[4]. Let $l_j$ denote the time for a server $j$ to process all remaining workloads, which is also the potential queuing time for new-coming tasks, $l_j(t) = \frac{X_j(t-1) + \sum_{i=1}^M w_i(t)\alpha_{ij}(t)}{v_j} = \frac{\sum_{i=1}^M X_{ij}(t-1) + w_i(t)\alpha_{ij}(t)}{v_j} = \sum_{i=1}^M l_{ij}(t)$. Then transition from time step $t$ to time step $t + 1$ is given in Alg. 1. Reward: $r_i(t) = R(\boldsymbol{l}(t), a_i(t), \delta_i(t))$, where $R$ is the reward function, $\boldsymbol{l}(t) = \sum_{j=1}^N l_j(t) = \sum_{i=1}^M l_i(t)$ denotes the estimated remaining time to process on each server, and $\delta_i(t)$ is a random variable that makes the process stochastic.

**Definition 1.** *(Makespan) In the selfish load balancing problem, the makespan is defined as:*

$$MS = \max_j(l_j), l_j = \sum_i l_{ij} \tag{1}$$

The network load balancing problem is featured as multi-commodity flow problems and is NP-hard, which makes it hard to solve with trivial algorithmic solution within micro-second level [39]. This problem can be formulated as a constrained optimisation problem for minimizing the makespan over an horizon $t \in [H]$:

$$minimize \sum_{t=h}^H \max_j l_j(t) \tag{2}$$

$$s.t. \quad l_j(t) = \frac{\sum_{i=1}^M (X_{ij}(t-1) + w_i(t)\alpha_{ij}(t))}{v_j}, \quad \sum_{i=1}^M w_i(t) \le \sum_{j=1}^N v_j, \quad w_i, v_j \in (0, +\infty) \tag{3}$$

$$X_{ij}(T) = \sum_{t=0}^T \max\{0, w_i(t)\alpha_{ij}(t) - \frac{v_j}{M}\}, \quad \sum_{j=1}^N \alpha_{ij}(t) = 1, \quad \alpha_{ij} \in [0, 1] \tag{4}$$

In modern realistic network load balancing system, the arrival of network requests is usually unpredictable in both its arriving rate and the expected workload, which introduces large stochasticity into the problem. Moreover, due to the existence of noisy measurements and partial observations, the estimation of makespan can be inaccurate, which indicates the actual server load states or processing capacities are not correctly captured. Instant collisions of elephant workloads or bursts of mouse workloads often happen, which do not indicate server processing capacity thus misleading the observation. To solve this issue, we introduce *fairness* as an alternative of the original objective makespan. Specifically, makespan is estimated on a per-server level, while the estimation of fairness can be decomposed to the LB level, which allows evaluating the individual LB performance without general loss. This is more natural in load balancing system due to the partial observability of LBs.

### 3.2 Distribution Fairness

We mainly introduce two types of load balancing distribution fairness: (1) variance-based fairness (VBF) and (2) product-based fairness (PBF). It will be proved that optimization over either fairness will be sufficient but not necessary for minimising the makespan.

---

[4] $X_{ij}(T) = \sum_{t=0}^T \max\{0, w_i(t)\alpha_{ij}(t) - \frac{v_j}{M}\}$

**Definition 2.** *(Variance-based Fairness) For a vector of time to finish all remaining jobs $\boldsymbol{l} = [l_1, \ldots, l_N]$ on each server $j \in [N]$, let $\bar{\boldsymbol{l}}(t) = \frac{1}{N} \sum_{j=1}^{N} \sum_{i=1}^{M} l_{ij}(t)$, the variance-based fairness for workload distribution is just the negative sample variance of the job time, which is defined as:*

$$F(\boldsymbol{l}) = -\frac{1}{N} \sum_{j=1}^{N} \left( l_j(t) - \bar{\boldsymbol{l}}(t) \right)^2 = -\frac{1}{N} \sum_{j=1}^{N} l_j^2(t) + \bar{\boldsymbol{l}}^2(t). \tag{5}$$

*VBF defined per LB is: $F_i(\boldsymbol{l}_i) = -\frac{1}{N} \sum_{j=1}^{N} l_{ij}^2(t) + \bar{\boldsymbol{l}}_i^2(t)$, where $\bar{\boldsymbol{l}}_i(t) = \frac{1}{N} \sum_{j=1}^{N} l_{ij}(t)$.*

**Lemma 3.** *The VBF for load balancing system satisfies the following property:*

$$F_i^{\pi_i, -\pi_i}(\boldsymbol{l}_i) - F_i^{\tilde{\pi}_i, -\pi_i}(\tilde{\boldsymbol{l}}_i) = F^{\pi_i, -\pi_i}(\boldsymbol{l}) - F^{\tilde{\pi}_i, -\pi_i}(\tilde{\boldsymbol{l}}) \tag{6}$$

This property makes VBF a good choice for the reward function in load balancing tasks. We will see more discussions in later sections. Proof of the lemma is provided in Appendix B.1.

**Proposition 4.** *Maximising the VBF is sufficient for minimising the makespan, subjective to the load balancing problem constraints (Eq. (3) and (4)): $\max F(\boldsymbol{l}) \Rightarrow \min \max_j(l_j)$. This also holds for per-LB VBF as $\max F_i(\boldsymbol{l}_i) \Rightarrow \min \max_j(\boldsymbol{l}_i)$.*

**Definition 5.** *(Product-based Fairness [40]) For a vector of time to finish all remaining jobs $\boldsymbol{l} = [l_1, \ldots, l_N]$ on each server $j \in [N]$, the product-based fairness for workload distribution is defined as: $F(\boldsymbol{l}) = F([l_1, \ldots, l_N]) = \prod_{j \in [N]} \frac{l_j}{\max(\boldsymbol{l})}$. PBF defined per LB is: $F_i(\boldsymbol{l}_i) = F([l_{i1}, \ldots, l_{iN}]) = \prod_{j \in [N]} \frac{l_{ij}}{\max(\boldsymbol{l}_i)}$.*

**Proposition 6.** *Maximising the product-based fairness is sufficient for minimising the makespan, subjective to the load balancing problem constraints (Eq. (3) and (4)): $\max F(\boldsymbol{l}) \Rightarrow \min \max(\boldsymbol{l})$.*

Proofs of proposition 4 and 6 are in Appendix B.1 and B.2, respectively. From proposition 4 and 6, we know that the two types of fairness can serve as an effective alternative objective for optimising the makespan, which will be leveraged in our proposed MARL method as valid reward functions.

### 3.3 Game Theory Framework

Markov game is defined as $\mathcal{MG}(H, M, \mathcal{S}, \mathcal{A}_{\times M}, \mathbb{P}, r_{\times M})$, where $H$ is the horizon of the game, $M$ is the number of player in the game, $\mathcal{S}$ is the state space, $\mathcal{A}_{\times M}$ is the joint action space of all players, $\mathcal{A}_i$ is the action space of player $i$, $\mathbb{P} = \{\mathbb{P}_h\}, h \in [H]$ is a collection of transition probability matrices $\mathbb{P}_h : \mathcal{S} \times \mathcal{A}_{\times M} \to \Pr(\mathcal{S})$, $r_{\times M} = \{r_i | i \in [M]\}, r_i : \mathcal{S} \times \mathcal{A}_{\times M} \to \mathbb{R}$ is the reward function for $i$-th player given the joint actions. The stochastic policy space for the $i$-th player in $\mathcal{MG}$ is defined as $\Pi_i : \mathcal{S} \to \Pr(\mathcal{A}_i), \Pi = \{\Pi_i\}, i \in [M]$.

For the Markov game $\mathcal{MG}$, the state value function $V_{i,h}^{\boldsymbol{\pi}} : \mathcal{S} \to \mathbb{R}$ and state-action value function $Q_{i,h}^{\boldsymbol{\pi}} : \mathcal{S} \times \mathcal{A} \to \mathbb{R}$ for the $i$-th player at step $h$ under policy $\boldsymbol{\pi} \in \Pi_{\times M}$ is defined as:

$$V_{i,h}^{\boldsymbol{\pi}}(s) := \mathbb{E}_{\boldsymbol{\pi}, \mathbb{P}} \left[ \sum_{h'=h}^{H} r_{i,h'}(s_{h'}, \boldsymbol{a}_{h'}) \middle| s_h = s \right], Q_{i,h}^{\boldsymbol{\pi}}(s, \boldsymbol{a}) := \mathbb{E}_{\boldsymbol{\pi}, \mathbb{P}} \left[ \sum_{h'=h}^{H} r_{i,h'}(s_{h'}, \boldsymbol{a}_{h'}) \middle| s_h = s, a_h = \boldsymbol{a} \right]. \tag{7}$$

**Definition 7.** *($\epsilon$-approximate Nash equilibrium) Given a Markov game $\mathcal{MG}(H, M, \mathcal{S}, \mathcal{A}_{\times M}, \mathbb{P}, \Pi_{\times M}, r_{\times M})$, let $\pi_{-i}$ be the policies of the players except for the $i$-th player, the policies $(\pi_i^*, \pi_{-i}^*)$ is an $\epsilon$-Nash equilibrium if $\forall i \in [M], \exists \epsilon > 0$,*

$$V_i^{\pi_i^*, \pi_{-i}^*}(s) \geq V_i^{\pi_i, \pi_{-i}^*}(s) - \epsilon, \forall \pi_i \in \Pi_i. \tag{8}$$

*If $\epsilon = 0$, it is an exact Nash equilibrium.*

**Definition 8.** *(Markov Potential Game) A Markov game $\mathcal{M}(H, M, \mathcal{S}, \mathcal{A}_{\times M}, \mathbb{P}, \Pi_{\times M}, r_{\times M})$ is a Markov potential game (MPG) if $\forall i \in [M], \pi_i, \tilde{\pi}_i \in \Pi_i, \pi_{-i} \in \Pi_{-i}, s \in \mathcal{S}$,*

$$V_i^{\pi_i, \pi_{-i}}(s) - V_i^{\tilde{\pi}_i, \pi_{-i}}(s) = \phi^{\pi_i, \pi_{-i}}(s) - \phi^{\tilde{\pi}_i, \pi_{-i}}(s), \tag{9}$$

*where $\phi(\cdot)$ is the potential function independent of the player index.*

**Lemma 9.** *Pure NE (PNE) always exists for PG, local maximisers of potential function are PNE. PNE also exists for MPG. [22]*

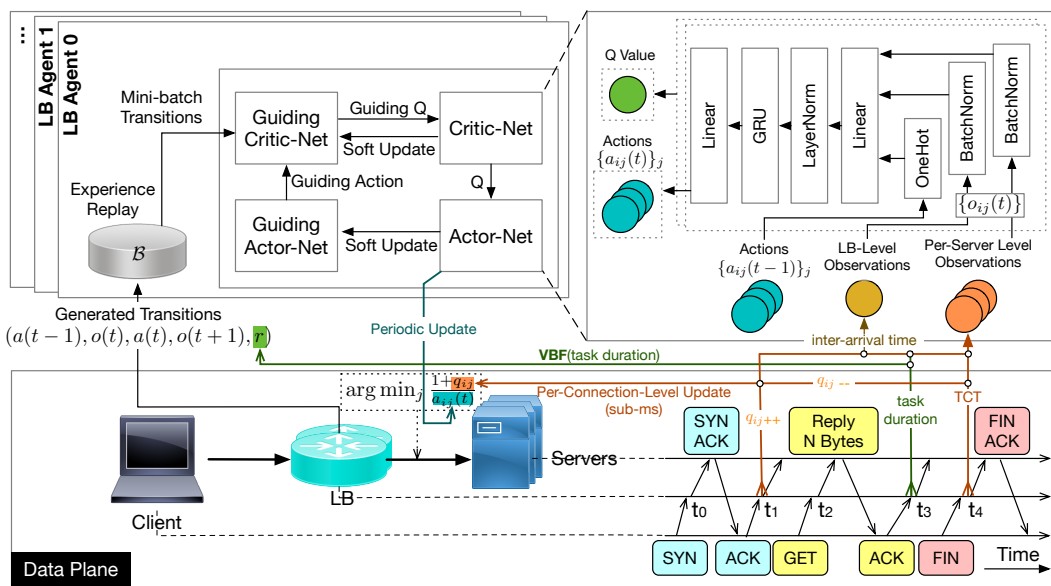

Figure 3: Overview of the proposed distributed MARL framework for network LB.

**Theorem 10.** *Multi-agent load balancing is MPG with the VBF $F_i(l_i)$ as the reward $r_i$ for each LB agent $i \in [M]$, then suppose for $\forall s \in \mathcal{S}$ at step $h \in [H]$, the potential function is time-cumulative total fairness: $\phi^{\pi_i,-\pi_i}(s) = \sum_{t=h}^{H} F^{\pi_i,-\pi_i}(l(t))$.*

The proof of the theorem is based on Lemma 3, and it's provided in Appendix B.3. This theorem is essential for establishing our method, since it proves that multi-agent load balancing problem can be formulated as a MPG with the time-cumulative VBF as its potential function. Also, the choice of per-LB VBF as reward function for individual agent is critical for making it MPG, it is easy to verify that PBF cannot guarantee such property. From Lemma 9 we know the maximiser of potential function is the NE of MPG, and from proposition 4 it is known that maximising the VBF gives the sufficient condition for minimising the makespan. Therefore, an effective independent optimisation with respect to the individual reward function specified in the above theorem will lead the minimiser of makespan for load balancing tasks. The effective independent optimisation here means the NE of MPG is achieved.

### 3.4 Distributed LB Method

With the above analysis, the load balancing problem can be formulated as an episodic version of multi-player partially observable Markov game, which we denote as $\mathcal{POMG}(H, M, \mathcal{S}, \mathcal{O}_{\times M}, \mathbb{O}_{\times M}, \mathcal{A}_{\times M}, \mathbb{P}, r_{\times M})$, where $M, H, \mathcal{S}, \mathcal{A}_{\times M}$ and $\mathbb{P}$ follow the same definitions as in Markov game $\mathcal{MG}$, $\mathcal{O}_{\times M}$ contains the observation space $O_i$ for each player, $\mathbb{O} = \{\mathbb{O}_h\}, h \in [H]$ is a collection of observation emission matrices, $\mathbb{O}_{i,h} : \mathcal{S} \rightarrow \Pr(\mathcal{O}_i)$, $r_{\times M} = \{r_i | i \in [M]\}, r_i : \mathcal{O}_i \times \mathcal{A}_{\times M} \rightarrow \mathbb{R}$ is the reward function for $i$-th LB agent given the joint actions. The stochastic policy space for the $i$-th agent in $\mathcal{POMG}$ is defined as $\Pi_i : \mathcal{O}_i \rightarrow \Pr(\mathcal{A}_i)$. As discussed in Sec. 2, the partial observability comes from the fundamental configuration of network LBs in DC networks, which allows LBs to observe only a partial of network traffic and does not give LBs information about the tasks (*e.g.* expected workload) distributed from each LB. The reward functions in our experiments are variants of distribution fairness introduced in Sec. 3.2. The potential functions can be defined accordingly based on the two fairness indices. The overview of the proposed distributed MARL framework is shown in Fig. 3.

In MPG, independent policy gradient allows finding the maximum of the potential function, which is the PNE for the game. This inspires us to leverage the policy optimisation in a decomposed manner, *i.e.*, distributed RL for policy learning of each LB agent. However, due to the partial observability of the system and the challenge of directly estimating the makespan (Eq. (1)), each agent cannot have a direct access to the global potential function. To address this problem, the aforementioned fairness

---

**Algorithm 2** Distributed LB for MPG

---

1: **Initialise:**
2:     LB policy $\pi_{\theta_i}$ and critic $Q_{\phi_i}$ networks, replay buffer $\mathcal{B}_i, \forall i \in [M]$;
3:     server processing speed function $v_j, \forall j \in [N]$;
4:     initial observed instant queue length on server $j$ by the $i$-th LB: $q_{ij} = 0, \forall i \in [M], j \in [N]$.
5: **while** not converge **do**
6:     Reset server load state $X_j(1) \leftarrow 0, \forall j \in [N]$
7:     Each LB agent $i$ ($i \in [M]$) receives individual observation $o_i(1)$
8:     **for** $t = 1, \ldots, H$ **do**
9:         Initialise distributed workload $m_{ij}, w_i(t) \leftarrow 0, i \in [M], j \in [N]$
10:         Get actions $a_i(t) \leftarrow \{a_{ij}(t)\}_{j=1}^N = \pi_{\theta_i}(o_i(t)), i \in [M]$
11:         **for** job $\tilde{w}$ arrived at LB $i$ between timestep $[t, t+1)$ **do**
12:             LB $i$ assigns $\tilde{w}$ to server $j = \arg\min_{k \in [N]} \frac{q_{ik}(t)+1}{a_{ik}(t)}$
13:             $m_{ij} \leftarrow m_{ij} + \tilde{w}, w_i(t) \leftarrow w_i(t) + \tilde{w}$
14:             $\alpha_{ij}(t) \leftarrow \frac{m_{ij}}{w_i(t)}$
15:         **end for**
16:         **for** each server $j$ **do**
17:             Update workload: $X_{ij}(t+1) \leftarrow \max\{X_{ij}(t) + w_i(t)\alpha_{ij}(t) - \frac{v_j}{M}, 0\}$
18:             $X_j(t+1) \leftarrow \sum_{i=1}^M X_{ij}(t)$
19:         **end for**
20:         Each agent receives individual reward $r_i(t)$
21:         Each agent $i$ collects observation $o_i(t+1), i \in [M]$
22:         Update replay buffer: $\mathcal{B}_i = \mathcal{B}_i \bigcup (a_i(t-1), o_i(t), a_i(t), r_i(t), o_i(t+1)), i \in [M]$
23:     **end for**
24:     Update critics with gradients: $\nabla_{\phi_i}\mathbb{E}_{(o_i,a_i,r_i,o_i') \sim \mathcal{B}_i}\left[\left(Q_{\phi_i}(o_i, a_i) - r_i - \gamma V_{\tilde{\phi}_i}(o_i')\right)^2\right]$
25:     where $V_{\tilde{\phi}_i}(o_i') = \mathbb{E}_{(o_i',a_i') \sim \mathcal{B}_i}[Q_{\tilde{\phi}_i}(o_i', a_i') - \alpha \log \pi_{\theta_i}(a_i'|o_i')], i \in [M]$
26:     Update policies with gradients: $-\nabla_{\theta_i}\mathbb{E}_{o_i \sim \mathcal{B}_i}[\mathbb{E}_{a \sim \pi_{\theta_i}}[\alpha \log \pi_{\theta_i}(a_i|o_i) - Q_{\phi_i}(o_i, a_i)]], i \in [M]$
27: **end while**
28: **return** final models of learning agents

---

(Sec. 3.2) can be deployed as the reward function for each agent, which makes the value function as a valid alternative for the potential function as an objective. This also transforms the joint objective (makespan or potential) to individual objectives (per LB fairness) for each agent. Proposition 4 and 6 verify that optimising towards these fairness indices is sufficient for minimising the makespan.

Alg. 2 shows the proposed distributed LB for load balancing problem, which is a partially observable MPG. The distributed policy optimisation is based on Soft Actor-Critic (SAC) [41] algorithm, which is a type of maximum-entropy RL method. It optimises the objective $\mathbb{E}[\sum_t \gamma^t r_t + \alpha \mathcal{H}(\pi_\theta)]$, whereas $\mathcal{H}(\cdot)$ is the entropy of the policy $\pi_\theta$. Specifically, the critic $Q$ network is updated with gradient $\nabla_\phi \mathbb{E}_{o,a}\left[\left(Q_\phi(o,a) - r(o,a) - \gamma \mathbb{E}_{o'}[V_{\tilde{\phi}}(o')]\right)^2\right]$, where $V_{\tilde{\phi}}(o') = \mathbb{E}_{a'}[Q_{\tilde{\phi}}(o', a') - \alpha \log \pi_\theta(a'|o')]$ and $Q_{\tilde{\phi}}$ is the target $Q$ network; the actor policy $\pi_\theta$ is updated with the gradient $\nabla_\theta \mathbb{E}_o[\mathbb{E}_{a \sim \pi_\theta}[\alpha \log \pi_\theta(a|o) - Q_\phi(o,a)]]$. Other key elements of RL methods involve the observation, action and reward function, which are detailed as following.

**Observation.** Each LB agent partially observes over the traffic that traverses through itself, including per-server-level and LB-level measurements. For each LB, per-server-level observations consist of – for each server – the number of on going tasks, and sampled task duration and task completion time (TCT). Specifically, in Alg. 2 line 12-14, $w_i$ is the coming workload on servers assigned by $i$-th LB, and it is not observable for LB. $q_{ik} + 1$ is the locally observed number of tasks on $k$-th server by $i$-th LB, due to the real-world constraints of limited observability at the Transport layer. The "+1" is for taking into account the new-coming task. Observations of task duration and TCT samples, along with LB-level measurements which sample the task inter-arrival time as an indication of overall system load state, are reduced to 5 scalars – *i.e.* average, 90th-percentile, standard deviation, discounted average and weighted discounted average[5] – as inputs for LB agents.

---

[5]Discounted average weights are computed as $0.9^{t'-t}$, where $t$ is the sample timestamp and $t'$ is the moment of calculating the reduced scalar.

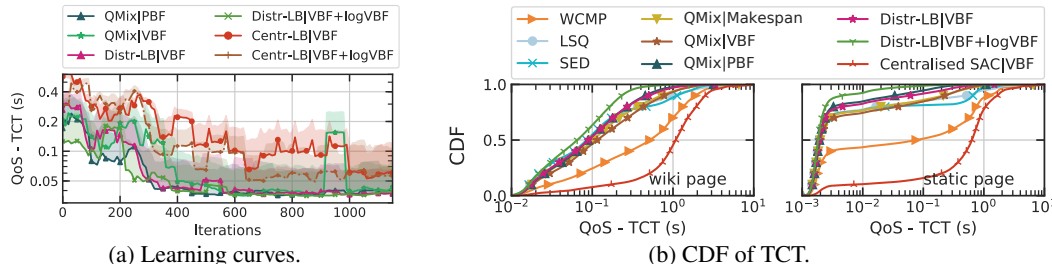

(a) Learning curves.      (b) CDF of TCT.

Figure 4: Experimental results show that the proposed distributed RL framework using proposed VBF as rewards converges and effectively achieves better load balancing performance (lower TCT and better QoS) than existing LB algorithms and CTDE RL algorithms.

**Action.** To bridge the different timing constraints between the control plane and data plane, each LB agent assigns the $j$-th server to newly arrived tasks using the ratio of two factors, $\arg\min_{k\in[N]} \frac{q_{ik}+1}{a_{ik}}$, where the number of on-going tasks $q_{ik}$ helps track dynamic system server occupation at per-connection level – which allows making load balancing decision at $\mu$s-level speed – and $a_{ik}$ is the periodically updated RL-inferred server processing speed. As in line 14 of Alg. 2, $\alpha_{ij}(t)$ is a statistical estimation of workload assignment distribution at time interval $[t, t+1)$.

**Reward.** The individual reward for distributed MPG LB is chosen as the VBF (as Def. 2) of the discounted average of sampled task duration measured on each LB agent, such that the LB group jointly optimise towards the potential function defined in Eq. (10). Task duration information is gathered as the time interval between the end of connection initialisation (*e.g.* 3-way handshake for TCP traffic) and the acknowledgement to the first data packet (*e.g.* the first ACK packet for TCP traffic). Given the limited and partial observability of LB agents, task duration information approximates the remaining workload $l$ by measuring the queuing and processing delay for new-coming tasks on each server. This PBF- and MS-based rewards are also implemented for CTDE MARL algorithm as a comparison.

**Model.** The architecture of the proposed RL framework is depicted in Fig. 3. Each LB agent consists of a replay buffer, and a pair of actor-critic networks, whose architecture is depicted on the top right. There is also a pair of guiding actor-critic networks, with the same network architectures but updated in a delayed and soft manner. Each LB agent takes observations $o_i(t)$ extracted from the data plane (*e.g.* numbers of ongoing tasks $\{q_{ij}\}$, task duration, TCT) and actions from previous timestep $a_i(t-1)$ as inputs, and periodically generates new actions $a_i(t)$, which is used to update the server assignment function $\arg\min_{j\in[N]} \frac{q_{ij}+1}{a_{ij}}$ in the data plane. The gated recurrent units (GRU) [42] are applied for all agents to leverage the sequential history information for handling partial observability.

## 4 Evaluation

We developed (i) an event-based simulator (App. C.1) to study the distance between the NE achieved by the proposed algorithm and the NE achieved by the theoretical optimal load balancing policy (with perfect observation), and (ii) a realistic testbed (App. C.2) on physical servers in a DC network providing Apache web services, with real-world network traffic [43], to evaluate the real-world performance of the proposed algorithm, in comparison with in-production state-of-the-art LB [3].

**Moderate-Scale Real-World Testbed:** As depicted in Fig. 4a, in a moderate-scale real-world DC network setup with 2 LB agents and 7 servers, after 120 episodes of training, the proposed distributed LB (Distr-LB) algorithm is able to learn from the environment based on VBF as rewards, and it converges to offer better QoS than QMix. Centralised RL agent (Centr-LB) has difficulties to learn within 120 episodes because of the increased state and action space. An empirical finding is that, by adding a log term to the VBF-based reward for Distr-LB, we help LB agents to become more sensitive to close-to-0 VBF during training ($\nabla_x \log f(x) > \nabla_x f(x)$ when $f(x) < 1$), therefore achieving better load balancing performance. As depicted in Fig. 4b, when comparing with in-production LB algorithms (WCMP, LSQ, SED), Distr-LB shows clear performance gains and reduced TCT for both types of web pages – Wikipedia pages require to query SQL databases thus they are more

Table 2: Comparison of average QoS (s) in moderate-scale real-world network setup.

| Method | | Period III (758.787 queries/s) | | Period IV (784.522 queries/s) | |
|---|---|---|---|---|---|
| | | Wiki | Static | Wiki | Static |
| WCMP | | $0.412 \pm 0.101$ | $0.134 \pm 0.059$ | $0.834 \pm 0.323$ | $0.492 \pm 0.276$ |
| LSQ | | $0.620 \pm 0.442$ | $0.339 \pm 0.316$ | $0.357 \pm 0.373$ | $0.173 \pm 0.299$ |
| SED | | $0.215 \pm 0.210$ | $0.051 \pm 0.081$ | $0.346 \pm 0.496$ | $0.169 \pm 0.330$ |
| **RLB-SAC** [40] | Jain | $0.193 \pm 0.073$ | $0.026 \pm 0.022$ | $0.204 \pm 0.084$ | $0.039 \pm 0.047$ |
| | G | $0.149 \pm 0.049$ | $0.015 \pm 0.011$ | $0.155 \pm 0.052$ | $0.011 \pm 0.011$ |
| **QMix-LB** | MS | $0.217 \pm 0.157$ | $0.048 \pm 0.069$ | $0.263 \pm 0.202$ | $0.073 \pm 0.092$ |
| | VBF | $0.141 \pm 0.025$ | $0.008 \pm 0.004$ | $0.286 \pm 0.162$ | $0.068 \pm 0.066$ |
| | PBF | $0.211 \pm 0.153$ | $0.047 \pm 0.078$ | $0.181 \pm 0.042$ | $0.018 \pm 0.009$ |
| **Distr-LB** | VBF | $0.159 \pm 0.054$ | $0.017 \pm 0.009$ | $0.196 \pm 0.091$ | $0.032 \pm 0.033$ |
| (this paper) | VBF+logVBF | $\mathbf{0.108 \pm 0.022}$ | $\mathbf{0.004 \pm 0.001}$ | $\mathbf{0.104 \pm 0.013}$ | $\mathbf{0.006 \pm 0.003}$ |
| **Centr-LB** | VBF | $1.068 \pm 0.386$ | $0.570 \pm 0.378$ | $1.378 \pm 0.377$ | $0.867 \pm 0.350$ |
| | VBF+logVBF | $0.759 \pm 0.254$ | $0.306 \pm 0.222$ | $1.013 \pm 0.168$ | $0.520 \pm 0.167$ |

Table 3: Comparison of average QoS (s) in moderate-scale simulator for different types of applications.

| | | 50%-CPU+50%-IO | 75%-CPU+25%-IO | 100%-CPU |
|---|---|---|---|---|
| Oracle | | $6.437 \pm 1.006$ | $1.469 \pm 0.102$ | $1.291 \pm 0.075$ |
| **QMix-LB** | PBF | $10.230 \pm 0.108$ | $1.828 \pm 0.054$ | $2.200 \pm 0.288$ |
| | VBF | $10.936 \pm 0.470$ | $2.023 \pm 0.255$ | $2.125 \pm 0.074$ |
| **Distr-LB** | VBF | $10.335 \pm 0.362$ | $\mathbf{1.695 \pm 0.104}$ | $\mathbf{1.643 \pm 0.016}$ |
| (this paper) | VBF+logVBF | $\mathbf{8.797 \pm 0.459}$ | $1.873 \pm 0.328$ | $2.004 \pm 0.042$ |

CPU-intensive, while static pages are IO-intensive. The comparison of average TCT using different LB algorithms is shown in Table 2 (99th percentile TCT in Table 12). The proposed Distr-LB also shows superior performance than the RL-based solution (RLB-SAC) [40] because of (i) a well designed MARL framework, and (ii) the use of recurrent neural network to handle load balancing problem as a sequential problem.

**NE Gap Evaluation with Simulation:** To evaluate the gap between the performance of Distr-LB and the theoretical optimal policy, we implement in the simulator an Oracle LB, which has perfect observation (inaccessible in real world) over the system and minimises makespan for each load balancing decision. Table 3 shows that, for different types of applications, Distr-LB is able to achieve closer-to-optimal performance than QMix. As the simulator is implemented based on the load balancing model formulated in this paper, our theoretical analysis can be directly applied, and VBF – as a potential function – helps independent cooperative LB agents to achieve good performance. The additional $log$ term shows empirical performance gains in real-world system, yet it is not necessarily the case in these simulation results. On one hand, the generated traffic of tasks in the simulation has higher expected workload ($> 1$s mean and stddev), while the $log$ terms is more sensitive to close-to-0 variances, which is the case in real-world experimental setups. On the other hand, though the simulator models the formulated LB problem, it fails to captures the complexity in the real-world system – *e.g.* Apache backlog, multi-processing optimisation, context switching, multi-level cache, network queues etc. For instance, batch processing [44] helps reduce cache and instruction misses, yet yields similar processing time for different tasks, thus the variance of task processing delay decreases and becomes closer to 0 in real-world system. The additional $log$ term exaggerates the low variance differences to better evaluate load balancing decisions. More detailed description about the simulator implementation can be found in App C.1 and ablation study on reward engineering is presented in App E.2.1.

Table 4: Comparison of average QoS (s) in large-scale real-world network setup.

| Method | | Period I (2022.855 queries/s) | | Period II (2071.129 queries/s) | |
|---|---|---|---|---|---|
| | | Wiki | Static | Wiki | Static |
| WCMP | | $0.473 \pm 0.102$ | $0.194 \pm 0.090$ | $0.460 \pm 0.241$ | $0.239 \pm 0.212$ |
| LSQ | | $0.266 \pm 0.127$ | $0.063 \pm 0.065$ | $0.218 \pm 0.246$ | $0.082 \pm 0.152$ |
| SED | | $0.169 \pm 0.062$ | $0.020 \pm 0.025$ | $0.166 \pm 0.141$ | $0.050 \pm 0.070$ |
| RLB-SAC-G [40] | | $0.182 \pm 0.049$ | $0.013 \pm 0.009$ | $0.111 \pm 0.029$ | $0.010 \pm 0.009$ |
| **QMix-LB** | VBF | $0.181 \pm 0.062$ | $0.019 \pm 0.020$ | $0.188 \pm 0.147$ | $0.052 \pm 0.075$ |
| | PBF | $0.210 \pm 0.041$ | $0.013 \pm 0.006$ | $0.104 \pm 0.009$ | $0.005 \pm 0.003$ |
| **Distr-LB** | VBF | $0.228 \pm 0.055$ | $0.019 \pm 0.011$ | $0.174 \pm 0.102$ | $0.035 \pm 0.039$ |
| (this paper) | VBF+logVBF | $\mathbf{0.161 \pm 0.033}$ | $\mathbf{0.008 \pm 0.003}$ | $\mathbf{0.094 \pm 0.015}$ | $\mathbf{0.004 \pm 0.001}$ |

Table 5: Comparison of 99-th percentile QoS (s) of Wiki pages under different traffic rates using large-scale real-world setup.

| Method | | Traffic Rate (queries/s) | | | | | | | | |
|---|---|---|---|---|---|---|---|---|---|---|
| | | 731.534 | 1097.3 | 1463.067 | 1828.834 | 2194.601 | 2377.484 | 2560.368 | 2743.251 | 2926.135 |
| LSQ | | 0.175 ±0.015 | 0.212 ±0.025 | 0.249 ±0.043 | 0.342 ±0.121 | 0.827 ±0.572 | 2.103 ±0.654 | 10.662 ±2.557 | 17.656 ±0.714 | 17.999 ±0.253 |
| SED | | 0.201 ±0.022 | 0.261 ±0.079 | 0.322 ±0.099 | 0.360 ±0.088 | 0.618 ±0.268 | 2.175 ±1.328 | 11.444 ±3.861 | 22.086 ±4.892 | 22.727 ±5.632 |
| **Distr-LB** (this paper) | VBF | **0.160** **±0.010** | **0.205** **±0.036** | **0.248** **±0.086** | **0.284** **±0.113** | 0.567 ±0.306 | **1.276** **±0.647** | 7.005 ±1.147 | 10.560 ±1.042 | 15.745 ±0.254 |
| | VBF+logVBF | 0.161 ±0.008 | 0.216 ±0.052 | 0.249 ±0.068 | 0.348 ±0.122 | **0.439** **±0.121** | 1.533 ±0.670 | **4.427** **±0.443** | **9.391** **±0.329** | **15.347** **±0.572** |

Table 6: Comparison of 99-th percentile QoS (s) of static pages under different traffic rates using large-scale real-world setup.

| Method | | Traffic Rate (queries/s) | | | | | | | | |
|---|---|---|---|---|---|---|---|---|---|---|
| | | 731.534 | 1097.3 | 1463.067 | 1828.834 | 2194.601 | 2377.484 | 2560.368 | 2743.251 | 2926.135 |
| LSQ | | 0.014 ±0.001 | 0.015 ±0.000 | 0.015 ±0.000 | 0.018 ±0.003 | 0.217 ±0.305 | 0.856 ±0.554 | 11.066 ±3.095 | 16.874 ±0.391 | 17.155 ±0.217 |
| SED | | 0.014 ±0.000 | 0.015 ±0.000 | 0.016 ±0.001 | 0.018 ±0.001 | 0.071 ±0.066 | 1.252 ±1.489 | 11.272 ±3.975 | 21.941 ±5.970 | 20.708 ±5.423 |
| **Distr-LB** (this paper) | VBF | 0.014 ±0.000 | 0.015 ±0.000 | 0.016 ±0.001 | **0.017** **±0.000** | **0.041** **±0.025** | **0.338** **±0.364** | 6.670 ±1.152 | 9.743 ±0.863 | 15.506 ±0.056 |
| | VBF+logVBF | 0.014 ±0.000 | 0.015 ±0.001 | 0.016 ±0.000 | 0.018 ±0.002 | 0.072 ±0.087 | 0.465 ±0.403 | **3.970** **±0.545** | **8.782** **±0.187** | **15.095** **±0.497** |

**Large-Scale Real-World Testbed:** To evaluate the performance of Distr-LB in large-scale DC networks in real world, we scale up the real-world testbed to have 6 LB agents and 20 servers and apply heavier network traffic ($> 2000$ queries/s) to evaluate the performance of the LB algorithms that achieved the best performance in moderate scale setups, in comparison with in-production LB algorithms. The test results after 200 episodes of training are shown in Table 4, where Distr-LB achieves the best performance in all cases. QMix also outperforms in-production LB algorithms. But as a CTDE algorithm, similar to the Centr-LB, it requires agents to communicate their trajectories, which – after 200 episodes of training – become 221MiB communication overhead at the end of each episode (episodic training), whereas 95%-percentile per-destination-rack flow rate is less than 1MiB/s [45].

**Scaling Experiments:** Using the same large-scale real-world testbed with 6 LB agents and 20 servers, we conduct scaling experiments by applying network traces with different traffic rates, comparing 4 LB methods with the best performances. The 99-th percentile QoS for both Wiki and static pages are shown in Table 5, 6. As listed in Table 5 and 6, under low traffic rates, when servers are all under utilised, the advantage of our proposed Distr-LB is not obvious because all resources are over-provisioned. With the increase of traffic rates (till servers are 100% saturated), our methods outperforms the best classical LB methods. More in-depth discussion and analaysis over the average job completion time for both types of pages in these scaling experiments are shown in Table 14 and 15 in App. E.2.2).

More details regarding the real-world DC testbed implementation is in App. C.2, training details are in App. D, complete evaluation results (both moderate-scale and large-scale) are in App. E and ablation studies – *e.g.* communication overhead of CTDE and centralised RL in real-world system, robustness of MARL algorithms in dynamic DC network environments – can be found in App. E.2.

## 5  Conclusion and Future Work

This paper proposes a distributed MARL approach for multi-agent load balancing problem, based on Markov potential game formulation. The proposed variance-based fairness for individual LB agent is critical for this formulation. Through this setting, the redundant communication overhead among LB agents is removed, thus improving the overall training and deployment efficiency in real-world systems, with the local observations only. Under such formulation, the effectiveness of our proposed distributed LB algorithm together with the proposed fairness are both theoretically justified and experimentally verified. It demonstrates a performance gain over another commonly applied fairness as well as centralised training methods like QMIX or centralised RL agent, in both simulation and real-world tests with different scales.

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
