# Appendix

## A    A Stochastic Markov Model of a $2$-Server Load Balancing Problem

The simulation results of Fig. 2b is based on a basic load balancing setup of 2 servers with different processing capacities $\frac{v_1}{v_2} = 2$ (*i.e.* server 1 is 2x faster than server 2). Each server has a queue of size $Q$, such that $0 \leq l_1, l_2 \leq Q$. Traffic arrivals and departures are modeled as Poisson processes with rates $\lambda$ (observed traffic), $\gamma$ (unobserved traffic), and $v_1, v_2$. With sufficiently short timeslots, it can be assumed that only one arrival or departure (at most) happen at a given timeslot (*i.e.* $\sum_{i=1}^{2}(\lambda_i + \gamma_i + v_i) \leq 1$); the system is then Markovian with the state $(l_1, l_2)$, departure rates $(\mu_1, \mu_2)$, and arrival rates $(\lambda_1, \lambda_2, \gamma_1, \gamma_2)$. For simplicity and stability, the system works at *nominal* capacity (*i.e.* $\lambda + \gamma = v$). With $q_i(n)_{l_i}$ denoting the probability (or probability density function), of server $q_i$ to have a queue length of $l_i$ at time-step $n$, the transition of server occupations between two time-steps can be described as, for $0 < l_i < Q$ (corner cases are treated accordingly):

$$q_i(n)_{l_i} - q_i(n-1)_{l_i} = (\lambda_i + \gamma_i) \cdot q_i(n-1)_{l_i-1} + v_i \cdot s_i(n-1)_{l_i+1} - (\lambda_i + \gamma_i + v_i) \cdot q_i(n-1)_{l_i}.$$

The QoS performance of each load balancing algorithm in Fig. 2b is measured as the weighted service duration of a connection ($\sum_{i\in\{1,2\}} \frac{l_i}{l_1+l_2} \frac{l_i}{\mu_i}$), under different configurations. When the LB has accurate observations and configurations (observing $100\%$ traffic – *i.e.* $\gamma = 0$ – and assigning server weights based on actual processing speeds $\frac{w_1}{w_2} = \frac{v_1}{v_2} = 2$), WCMP and SED have the best performance. When the LB observes only partial network traffic ($50\% - Q$ and $33\% - Q$ corresponds to $\gamma = \lambda$, $\gamma = 2*\lambda$, respectively) and the rest of the network traffic is uniformly split between the two servers ($\gamma_1 = \gamma_2$), LSQ and SED outperform WCMP, which is agnostic to instant server occupancy. However, partial traffic observation also degrades the performance of LSQ and SED. When LBs have inaccurate server weights ($\sim W$ *i.e.* in case of mis-configuration, $\frac{w_1}{w_2} = \frac{1}{2}$, while $\frac{\mu_1}{\mu_2} = 2$), WCMP and SED exhibit degraded performance even when the LB agent sees all the traffic ($\gamma = 0$). Taking both server queue lengths and processing speeds into account, SED makes more informed load balancing decisions, yet its performance risks being degraded by both partial observations on server queue lengths and inaccurate server weights.

## B    Analysis of Distribution Fairness

### B.1    Analysis of VBF

**Lemma 11.** *The VBF for load balancing system satisfies the following property:*

$$F_i^{\pi_i, -\pi_i}(\boldsymbol{l}_i) - F_i^{\tilde{\pi}_i, -\pi_i}(\tilde{\boldsymbol{l}}_i) = F^{\pi_i, -\pi_i}(\boldsymbol{l}) - F^{\tilde{\pi}_i, -\pi_i}(\tilde{\boldsymbol{l}}) \tag{10}$$

*Proof.* From the definition of the variance-based fairness (as Def. 2) we have the following for $\forall i \in [M], j \in [N]$,

$$F^{\pi_i, -\pi_i}(\boldsymbol{l}) = -\frac{1}{N} \sum_{j=1}^{N}(l_j - \bar{\boldsymbol{l}})^2 \tag{11}$$

$$F_i^{\pi_i, -\pi_i}(\boldsymbol{l}_i) = -\frac{1}{N} \sum_{j=1}^{N}(l_{ij} - \bar{l}_i)^2 \quad (\bar{l}_i = \frac{1}{N}\sum_{j=1}^{N} l_{ij}) \tag{12}$$

By indexing the agent $i$ as the one to change its strategy and slightly abusing notation, denote $l_j = l_{ij} + l_{-ij}$, where $l_{-ij} = \sum_{k \neq i} l_{kj}$.

$$F^{\pi_i, -\pi_i}(\boldsymbol{l}) = -\frac{1}{N} \sum_{j=1}^{N} (l_{ij} + l_{-ij} - \overline{(l_i + l_{-i})})^2 \quad \text{(where } \overline{(l_i + l_{-i})} = \frac{1}{N} \sum_j (l_{ij} + l_{-ij})) \tag{13}$$

$$= -\frac{1}{N} \sum_{j=1}^{N} [l_{ij} + l_{-ij} - (\bar{l}_i + \bar{l}_{-i})]^2 \tag{14}$$

$$= -\frac{1}{N} \sum_{j=1}^{N} [(l_{ij} - \bar{l}_i)^2 + (l_{-ij} - \bar{l}_{-i})^2 - 2(l_{ij} - \bar{l}_i)(l_{-ij} - \bar{l}_{-i})] \tag{15}$$

$$= -\frac{1}{N} \sum_{j=1}^{N} (l_{ij} - \bar{l}_i)^2 - \frac{1}{N} \sum_{j=1}^{N} [(l_{-ij} - \bar{l}_{-i})^2 - \frac{2}{N} \sum_{j=1}^{N} (l_{ij} - \bar{l}_i)(l_{-ij} - \bar{l}_{-i})] \tag{16}$$

$$= F_i^{\pi_i, -\pi_i}(\boldsymbol{l}_i) - \frac{1}{N} \sum_{j=1}^{N} (l_{-ij} - \bar{l}_{-i})^2 \quad (\sum_{j=1}^{N} (l_{ij} - \bar{l}_i) = 0) \tag{17}$$

where the second term is a common term not depend on the changing policy $\pi_i$. Therefore, the second term will be cancelled in $F^{\pi_i, -\pi_i}(\boldsymbol{l}) - F^{\tilde{\pi}_i, -\pi_i}(\tilde{\boldsymbol{l}}) = F_i^{\pi_i, -\pi_i}(\boldsymbol{l}_i) - F_i^{\tilde{\pi}_i, -\pi_i}(\tilde{\boldsymbol{l}}_i)$, thus finishes the proof. $\qquad \square$

**Proposition 12.** *Maximising the VBF is sufficient for minimising the makespan, subjective to the load balancing problem constraints (Eq. (3) and (4)):*

$$\max F(\boldsymbol{l}) \Rightarrow \min \max_j (l_j) \tag{18}$$

*this also holds for per-LB VBF as $\max F_i(\boldsymbol{l}_i) \Rightarrow \min \max_j(\boldsymbol{l}_i)$.*

*Proof.* Given the stability constraint in Eq. (3) $\sum_{i=1}^{M} w_i(t) \leq \sum_{j=1}^{N} v_j$, we denote the total amount of workload in the system $C = \sum_{j=1}^{N} l_j$, and $l_k = \max_{j \in [N]} l_j$. Based on the constraint in Eq. (4), we have $C \geq 0, l_j(t) \geq 0$.

$$\max F(\boldsymbol{l}) \Leftrightarrow \min -F(\boldsymbol{l}) \tag{19}$$

$$-F(\boldsymbol{l}) = \frac{1}{N} \sum_{j=1}^{N} ((l_j) - \bar{l})^2 \tag{20}$$

$$= \frac{1}{N} \sum_{j=1}^{N} (l_j - \frac{C}{N})^2 \tag{21}$$

$$= \frac{1}{N} \sum_{j=1}^{N} l_j^2 - \frac{2C}{N^2} \sum_{j=1}^{N} l_j + \frac{C^2}{N^2} \tag{22}$$

$$= \frac{1}{N} \sum_{j=1}^{N} l_j^2 - \frac{C^2}{N^2} \tag{23}$$

$$\leq [(\max_j l_j)^2 - \frac{C^2}{N^2}] \quad \text{(by means inequality)} \tag{24}$$

with the equivalence achieved when $l_j = l_k, \forall j \neq k, j \in [N]$ holds. Therefore,

$$\max F(\boldsymbol{l}) \Rightarrow \min (l_k)^2 - \frac{C^2}{N^2} \tag{25}$$

$$\Leftrightarrow \min l_k \tag{26}$$

$$\Leftrightarrow \min \max_{j \in [n]} l_j \tag{27}$$

and the condition is sufficient but not necessary because $\min (l_k)^2 - \frac{C^2}{N^2}$ is essentially minimizing the upper bound of $-F(\boldsymbol{l})$. $\qquad \square$

## B.2 Analysis of PBF

**Proposition 13.** *Maximising the product-based fairness is sufficient for minimising the makespan, subjective to the load balancing problem constraints (Eq. (3) and (4)):*

$$\max F(\boldsymbol{l}) \Rightarrow \min \max(\boldsymbol{l}) \tag{28}$$

*Proof.* For a vector of workloads $\boldsymbol{l} = [l_1, \ldots, l_N]$ on each server $j \in [N]$, by the definition of fairness,

$$\max F(\boldsymbol{l}) = \max \frac{\prod_{j \in [N]} l_j}{\max_{k' \in [N]} l_{k'}} \tag{29}$$

WLOG, let $l_k = \max_{k' \in [N]} l_{k'}$, then,

$$\max F(\boldsymbol{l}) = \max \prod_{j \in [N], j \neq k} l_j \tag{30}$$

Similar to the proof of Proposition 12, given the stability constraint in Eq. (3) $\sum_{i=1}^{M} w_i(t) \leq \sum_{j=1}^{N} v_j$, we denote the total amount of workload in the system $C = \sum_{j=1}^{N} l_j$. Based on the constraint in Eq. (4), we have $C \geq 0, l_j(t) \geq 0$. By means inequality,

$$\left( \prod_{j \in [N], j \neq k} l_j \right)^{\frac{1}{N-1}} \leq \frac{\sum_{j \in [N], j \neq k} l_j}{N - 1} = \frac{C - l_k}{N - 1}. \tag{31}$$

with the equivalence achieved when $l_i = l_j, \forall i, j \neq k, i, j \in [N]$ holds. Therefore,

$$\max F(\boldsymbol{l}) \Rightarrow \max \frac{C - l_k}{N - 1} \tag{32}$$

$$\Leftrightarrow \min l_k \tag{33}$$

$$\Leftrightarrow \min \max_{j \in [N]} l_j \tag{34}$$

The inverse may not hold since $\max \frac{C - l_k}{N-1}$ does not indicates $\max F(\boldsymbol{l})$, so maximising the linear product-based fairness is sufficient but not necessary for minimising the makespan. This finishes the proof. $\square$

## B.3 VBF for MPG

**Theorem 14.** *Multi-agent load balancing is MPG with the VBF $F_i(\boldsymbol{l}_i)$ as the reward $r_i$ for each LB agent $i \in [M]$, then suppose for $\forall s \in \mathcal{S}$ at step $h \in [H]$, the potential function is time-cumulative total fairness: $\phi^{\pi_i, -\pi_i}(s) = \sum_{t=h}^{H} F^{\pi_i, -\pi_i}(\boldsymbol{l}(t))$.*

*Proof.*

$$V_i^{\pi_i, \pi_{-i}}(s) - V_i^{\tilde{\pi}_i, \pi_{-i}}(s) = \mathbb{E}_{\pi_i, \pi_{-i}} \left[ \sum_{t=h}^{H} r_{i,t}(s_t, \boldsymbol{a}_t) \Big| s_h = s \right] - \mathbb{E}_{\tilde{\pi}_i, \pi_{-i}} \left[ \sum_{t=h}^{H} r_{i,t}(s_t, \tilde{a}_{i,t}, a_{-i,t}) \Big| s_h = s \right] \tag{35}$$

$$= \mathbb{E}_{\pi_i, \pi_{-i}} \left[ \sum_{t=h}^{H} F_i(\boldsymbol{l}_i(t)) \right] - \mathbb{E}_{\tilde{\pi}_i, \pi_{-i}} \left[ \sum_{t=h}^{H} F_i(\tilde{\boldsymbol{l}}_i(t)) \right] \tag{36}$$

$$= \sum_{t=h}^{H} \left( F^{\pi_i, -\pi_i}(\boldsymbol{l}) - F^{\tilde{\pi}_i, -\pi_i}(\tilde{\boldsymbol{l}}) \right) \quad \text{(Lemma 3)} \tag{37}$$

$$= \phi^{\pi_i, -\pi_i}(s) - \phi^{\tilde{\pi}_i, -\pi_i}(s) \tag{38}$$

Notice that $s$ is the ground truth state of the environment, therefore involving the expected time $\boldsymbol{l}$ to finish remaining jobs. $\square$

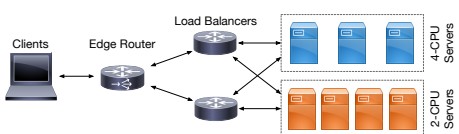
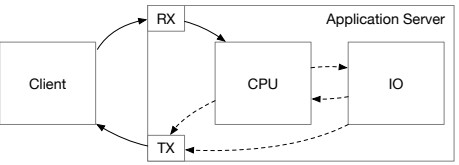

(a) An example of network topology with two groups of 7 servers.

(b) Illustration of the processing states of connection requests. Solid and dashed arrows represent deterministic and non-deterministic procedures respectively.

Figure 5: Simulator implementation details.

**Lemma 15.** *NE for MPG is $\epsilon$-approximate NE for $\epsilon$-approximate MPG.* [46]

*Proof.* We know NE $(\pi_i^*, \pi_{-i}^*)$ for MPG,

$$V_i^{\pi_i^*, \pi_{-i}^*}(s) - V_i^{\tilde{\pi}_i, \pi_{-i}^*}(s) = \phi^{\pi_i^*, \pi_{-i}^*}(s) - \phi^{\tilde{\pi}_i, \pi_{-i}^*}(s) \geq 0 \tag{39}$$

the policies can be $\epsilon$-approximate NE for another game with a different value function $\widehat{V}$ but the same potential function,

$$\widehat{V}_i^{\pi_i^*, \pi_{-i}^*}(s) - \widehat{V}_i^{\tilde{\pi}_i, \pi_{-i}^*}(s) \geq \epsilon, \forall i \in [N], \tilde{\pi}_i \in \Pi_i, s \in \mathcal{S} \tag{40}$$

thus,

$$\left| \left( \widehat{V}_i^{\pi_i^*, \pi_{-i}^*}(s) - \widehat{V}_i^{\tilde{\pi}_i, \pi_{-i}^*}(s) \right) - \left( \phi^{\pi^*, \pi_{-i}^*}(s) - \phi^{\tilde{\pi}, \pi_{-i}^*}(s) \right) \right| \leq \epsilon \tag{41}$$

which satisfies the definition of $\epsilon$-approximate MPG. $\qquad\square$

## C  Implementation

### C.1  Simulator

In order to compare the proposed RLB algorithms to the theoretically optimal solution which has perfect observation over the system – which is not achievable in real-world system, we implement an event-driven simulator to simulate the discrete process of network flow arrival and departure in a networked system. The simulator implements the network topology as in Fig. 5a, where each LB is connected to all servers.

Real-world network applications can be CPU-bound or IO-bound [47,48]. The simulator allows configuring applications that require multi-stage processes switching between CPU/IO queues (Fig. 5b). For instance, a connection request for a 2-stage application is first processed in the CPU queue, then in the IO queue, before being sent back to the client.

Two different processing models are used for CPU and IO queues, respectively. A FIFO model is defined for CPU queues, and connections that arrive when no CPU is available will be blocked in a backlog queue until there is an available CPU. Realistic network applications feature blocked processor sharing model [47,48], in which the instantaneous processing speed for each task $\hat{v}_j(t)$ at time $t$ on the $j$-th server is:

$$\hat{v}_j(t) = \begin{cases} 1 & |w_j(t)| \leq p_j, \\ \frac{p_j}{\min(\hat{p}_j, |w_j(t)|)} & |w_j(t)| > p_j, \end{cases} \tag{42}$$

where $|w_j(t)|$ denotes the number of on-going tasks, and $p_j$ denotes the number of processors on the $j$-th server. At any given moment, the maximum number of tasks that can be processed is $\hat{p}_j$. Tasks that arrive when $|w_j(t)| \geq \hat{p}_j$ will be blocked in a wait queue (similar to backlog in *e.g.* Apache) and will not be processed until there is an available slot in the CPU processing queue. However, this does not happen under the constraints in Eq. (3) as the task arrival rates are always slower than task departure rates (processing speed). The server processing speed therefore is $v_j(t) = \hat{v}_j(t)|w_j(t)|$. IO is simulated as a simple processor sharing model, in which the instantaneous processing speed is the inverse of the number of connections in the IO queue. The backlog queue length of each server is configured as 64. Connections that arrive when the backlog queues are full will be rejected, with 40s timeout. Communication latency between 2 nodes is uniformly distributed between 0.1ms and 1ms.

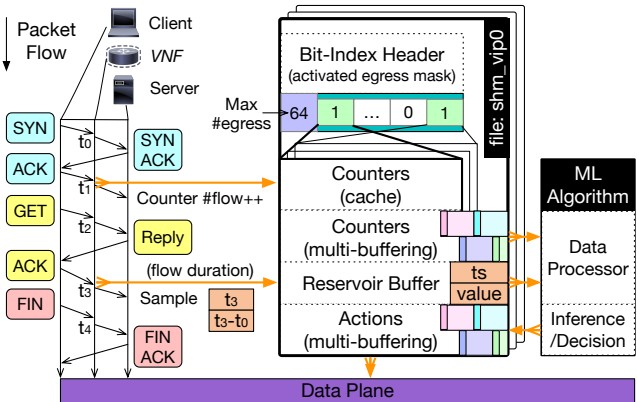

Figure 6: Feature collection mechanism: `shm` layout and data flow pipeline.

## C.2  Real-World DC Testbed

### C.2.1  System Platform

Application servers are virtualised on $4$ UCS B200 M4 servers, each with one Intel Xeon E5-2690 v3 processor ($12$ physical cores and $48$ logical cores), interconnected by UCS 6332 16UP fabric. Operating systems are `Ubuntu 18.04.3 LTS` (`GNU/Linux 4.15.0-128-generic x86_64`). Compilers are `gcc version 7.5.0` (`Ubuntu 7.5.0-3ubuntu1 18.04`). Applications employed in this paper are the following: `Apache 2.4.29`, `VPP v20.05`, `MySQL 5.7.25-0ubuntu0.18.04.2`, and `MediaWiki v1.30`. The VMs are deployed on the same layer-$2$ link, with statically configured routing tables. Apache HTTP servers share the same VIP address on one end of GRE tunnels with the load balancer on the other end.

### C.2.2  Apache HTTP Servers

The Apache servers use `mpm_prefork` module to boost performance. Each server has maximum $32$ worker threads and TCP backlog is set to $128$. In the Linux kernel, the `tcp_abort_on_overflow` parameter is enabled, so that a TCP RST will be triggered when the queue capacity of TCP connection backlog is exceeded, instead of silently dropping the packet and waiting for a SYN retransmit. With this configuration, the FCT measures application response delays rather than potential TCP SYN retransmit delays. Two metrics are gathered as ground truth server load state on the servers: CPU utilization and instant number of Apache busy threads. CPU utilization is calculated as the ratio of non-idle cpu time to total cpu time measured from the file `/proc/stat` and the number of Apache busy threads is assessed via Apache's *scoreboard* shared memory.

### C.2.3  $24$-Hour Wikipedia Replay Trace

To create Wikipedia server replicas, an instance of MediaWiki[6] of version $1.30$, a MySQL server and the `memcached` cache daemon are installed on each of the application server instance. *WikiLoader* tool [49] and a copy of the English version of Wikipedia database [43], are used to populate MySQL databases. The 24-hour trace is obtained from the authors of [43] and for privacy reasons, the trace does not contain any information that exposes user identities.

### C.2.4 Feature Collection and Policy Update in the Data Plane

---

**Algorithm 3** Reservoir sampling with no rejection

---

1: $k \leftarrow$ reservoir buffer size
2: $buf \leftarrow [(0,0), \ldots, (0,0)]$                     ▷ Size of $k$
3: **for** each observed sample $v$ arriving at $t$ **do**
4:     $randomId \leftarrow rand()$
5:     $idx \leftarrow randomId\%N$                 ▷ randomly select one index
6:     $buf[idx] \leftarrow (t,v)$                 ▷ register sample in buffer
7: **end for**

---

In order to apply RL in an asynchronous close-loop load balancing framework with high scalability and low latency, communication between the load balancer data plane and the ML application is implemented via POSIX shared memory (`shm`). This mechanism allows features to be extracted from the data plane and conveyed to the RL agent – with absolutely zero control message or communication overhead, and allows data-driven decisions generated by the RL agent to be updated asynchronously on the load balancer.

The pipeline of the data flow over the lifetime of a TCP connection is depicted in Fig. 6. By statefully tracking flow states, on receipt of different networking packets, we inspect packet header and collect networking features as counters or samples. Quantitative features (task duration and task completion time) are collected as samples, using reservoir sampling (Algorithm 3). Since networking environments are dynamic, it is important to capture not only the features, but also the sequential information of the system. Reservoir sampling gathers a representative group of samples in fix-sized buffer from a stream, with $\mathcal{O}(1)$ insertion time. It captures both the sampling timestamps and exponentially-distributed numbers of samples over a time window, which help conduct sequential pattern analysis[7]. For a Poisson stream of events with rate $\lambda$, the expectation of the amount of samples that are preserved in buffer after $n$ steps is $E = \lambda \left( \frac{k-1}{k} \right)^{\lambda n}$, where $k$ is the size of reservoir buffer. On the other hand, counters are collected atomically and made available to the data processing agent using multi-buffering.

Cloud services have different characteristics and they are identified by virtual IPs (VIPs), which correspond to clusters of provisioned resources – *e.g.* servers, identified by a unique direct IP (DIP). In production, cloud DCs are subject to high traffic rates and their environments and topologies change dynamically. This requires to organise collected features in a generic yet scalable format, and make features available for ML algorithms without disrupting the data plane. We organise observations of each VIPs in independent POSIX shared memory (`shm`) files, to provide scalable and dynamic service management. In each `shm` file, collected features are further partitioned by egress equipments so that spatial information can be distinguished, including counters and reservoir samples. Fig. 6 exemplifies the `shm` layout and data flow.

The bit-index binary header helps efficiently identify active application servers. Each server has its own independent memory space, storing counters, reservoir samples, and data plane policies (actions) if necessary. As depicted in Fig. 6, on receipt of the first `ACK` from the client to a specific server $i$, VNF increments the number of flows in the counters cache of node $i$ with $\mathcal{O}(1)$ complexity. With the same level of complexity, quatitative features (*e.g.* flow duration $t_3 - t_0$ gathered at $t_3$ in Fig. 6) can be stored in the reservoir buffer of node $i$ using Algorithm 3. Gathered features (counters and samples) are made available in an organised layout and they can be quickly accessed by ML algorithms running in a different process. With the bit-index header, locating features for a given server requires $\mathcal{O}(1)$ computational complexity and $\mathcal{O}(k)$ memory complexity, where $k$ is the reservoir buffer size. Obtained features for all active servers can then be aggregated and processed to make further inferences or data-driven decisions, which can be written back to the memory space of each server ($\mathcal{O}(1)$ computational complexity).

---

[6]https://www.mediawiki.org/wiki/Download

[7]Based on the characteristics of different system dynamics, *e.g.* long-term distribution shifts or short-term oscillations, tuning the reservoir sampling mechanism enables to collect different statistical representations of the states.

| Operation / Complexity | | Computation | Memory |
|---|---|---|---|
| Add / Remove VIP | | $\mathcal{O}(1)$ | $\mathcal{O}(kN + mN)$ |
| Add server | | $\mathcal{O}(1)$ | $\mathcal{O}(k + m)$ |
| Remove server | | $\mathcal{O}(1)$ | $\mathcal{O}(1)$ |
| Register reservoir sample Update counter (cache) | | $\mathcal{O}(1)$ | $\mathcal{O}(1)$ |
| Update counters / actions (multi-buffering) | | $\mathcal{O}(1)$ | $\mathcal{O}(N)$ |
| Get the latest observation | 1 node | $\mathcal{O}(m)$ | $\mathcal{O}(k + m)$ |
| | All nodes | | $\mathcal{O}(kN + mN)$ |
| Update action in the data plane | 1 node | $\mathcal{O}(m)$ | $\mathcal{O}(1)$ |
| | All nodes | | $\mathcal{O}(N)$ |

Table 7: Computation and memory complexity of different operations, where $k$ is the size of reservoir buffer, $N$ is the number of servers, and $m$ is the level of multi-buffering.

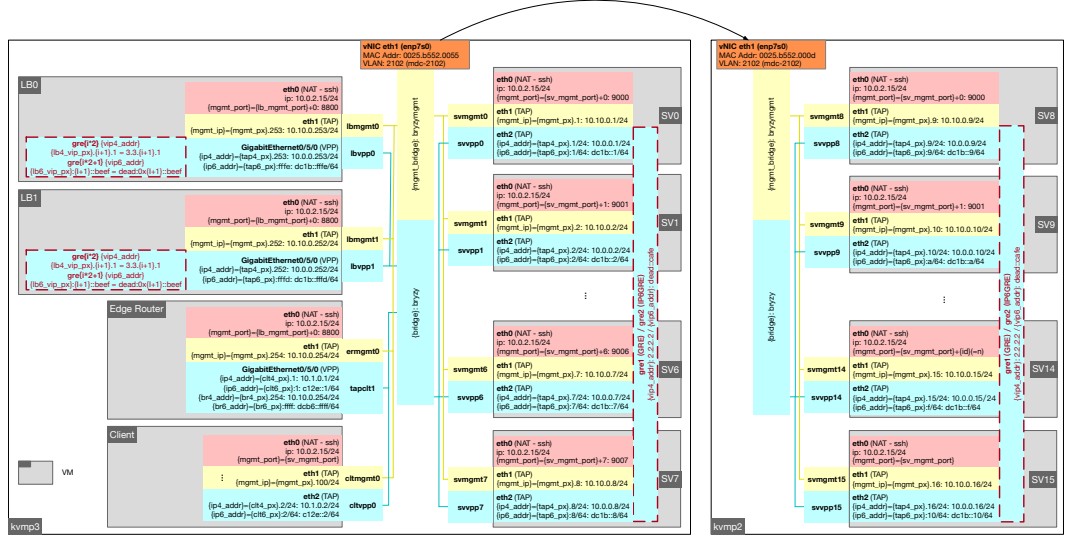

Figure 7: Network topology of the real-world DC testbed.

While quantitative features are collected using reservoir sampling, counters are incremented by the data plane in the cache, and periodically drawn from cache using $m$-level multi-buffering with incremental sequence ID. When copying data between cache and buffer, the sequence ID is set to $0$ to avoid I/O conflicts. Pulling the counters from cache to multi-buffering requires $\mathcal{O}(1)$ computational complexity and maximal $\mathcal{O}(N)$ memory complexity. ML algorithms can pull the latest observations from the multi-buffering with no disruption in the data plane, with $\mathcal{O}(m)$ computational complexity to find the buffer with the highest sequence ID. Similarly, new network policies and data-driven decisions (*e.g.* forwarding rules) can be updated to the data plane via action multi-buffering with $\mathcal{O}(m)$ computational complexity.

To summarise, both computation and memory space complexity is presented in Table 7. The whole dataflow is asynchronous and avoid stalling in the data exchange process in both the data plane and the control plane.

### C.2.5 Network Topology

For reproducibility, the network topology is depicted in Fig. 7. Two physical servers are connected via a VLAN. Each device is an instance of KVM, which is widely used for in-production vitualised Content Delivery Networks (CDNs).

Table 8: Survey on real-world testbed configurations.

| Related Work | Testbed Scale | Note |
|---|---|---|
| 6LB [4] | 2 LB + 28 servers (2-CPU each) | Our paper uses the same network trace as input traffic. |
| Ananta [50] | 14 LBs for 12 VIPs | The exact number of servers per VIP and the in-production traffic is not documented in the paper. |
| Beamer [51] | 2 LB + 8 servers (small) 4 LB + 10 servers (large) | Large scale experiments are conducted with 700 active HTTP connections max. |
| Duet [52] | 3 software LB + 3 hardware LB + 34 servers | Synthetic traffic is applied so that the server cluster behind VIP processes 60k (identical) packets per second. |
| SilkRoad [14] | 1 hardware LB or 3 software LB per VIP | Real-world PoP traffic is applied, where one server cluster behind VIP processes on average 309.84 active connections per second. |
| Cheetah [20] | 2 LB + 24 servers | A Python generator creates 1500-2500 synthetic requests/s. |

### C.2.6 Realistic Testbed

Modern data center may comprise thousands of servers and hundreds of LBs. However, each independent service is exposed in a modular way at one or several virtual IP (VIP) addresses to receive requests, running over a cluster of servers. Each server in the cluster can be identified by a unique direct IP (DIP). Traffic and queries from the clients destined to a VIP are load balanced among the DIPs of the service. The development of virtualization, where computers are emulated and/or sharing an isolated portion of the hardware by way of Virtual Machines (VMs), or run as isolated entities (containers) within the same operating system kernel, has accelerated the commoditization of compute resources. Therefore, the gigantic in-production data center network are typically partitioned into small pods, where different services (VIPs) are hosted. To justify the setups of our experiment satisfy the "real-world" requirement, we present a brief survey of real-world DC setup based on a set of state-of-the-art load balancing research papers, which are summarized below (Table 8).

Using 2 physical servers (48 CPUs each), we have made our best effort to find a configuration that allows us to conduct experiments similar to real-world setups. Based on the survey above, we believe that the experiments conducted in this paper have reasonable scale – not only in terms of number of agents (2/6 LBs) and servers (7/20 servers), but also in terms of traffic rates – more than 2k queries per second per VIP and more than 1150.76 concurrent connections in large scale experiments —- and are representative of real-world circumstances.

### C.3 Benchmark Load Balancing Methods

To compare load balancing performance, 4 state-of-the-art workload distribution algorithms are implemented. Equal-cost multi-path (ECMP) randomly assigns servers to tasks with a server assignment function $\mathbb{P}(j) = \frac{1}{n}$, where $\mathbb{P}(j)$ denotes the probability of assigning the $j$-th server [13]. Weighted-cost multi-path (WCMP) assigns servers based on their weights derived, and has an assignment function as $\mathbb{P}(j) = \frac{v_j}{\sum v_j}$ [3]. Local shortest queue (LSQ) assigns the server with the shortest queue, *i.e.* $\arg\min_{j \in [n]} |w^j(t)|$ [19]. Shortest expected delay (SED) assigns the server the shortest queue normalized by the number of processors, *i.e.* $\arg\min_{j \in [n]} \frac{|w^j(t)|+1}{v_j}$ [2], and is expected to have the best performance among conventional heuristics. In the simulator, an *Oracle* LB algorithm is implemented, which distributes connections to the server which is expected to finish all its job with the lowest delay (including the new connection). The Oracle LB is aware of the remaining time of each connection, which is otherwise not observable for network LBs in real-world setups. When receiving a new connection, the Oracle LB algorithm calculates the remaining time to process on each server (assuming the newly received connection is assigned on the server as well) and assigns the server with the lowest remaining time to process to the new-coming connection, to make sure that the makespan is always minimised with the global observation, which is not possible to be achieved in real-world system. The load balancing decisions for the Oracle algorithm are also made immediately for the Oracle LB algorithm.

Table 9: Hyperparameters in MARL-based LB.

| | Hyperparameter | Simulation | Experiments | |
| | | Moderate-Scale | Moderate-Scale | Large-Scale |
| --- | --- | --- | --- | --- |
| Distributed LB | Learning rate | $3 \times 10^{-4}$ | $1 \times 10^{-3}$ | $1 \times 10^{-3}$ |
| | Batch size | 25 | 25 | 12 |
| | Hidden dimension | 64 | 64 | 128 |
| | Hypernet dimension | 32 | 32 | 64 |
| | Replay buffer size | 3000 | 3000 | 3000 |
| | Episodes | 500 | 120 | 200 |
| | Updates per episode | 10 | 10 | 10 |
| | Step interval | 0.5s | 0.25s | 0.25s |
| | Target entropy | $-|\mathcal{A}|$ | $-|\mathcal{A}|$ | $-|\mathcal{A}|$ |
| LB System | TCT Distribution | Exponential | Real-world trace | Real-world trace |
| | Average TCT | 1s | 200ms | 200ms |
| | Average bytes per task | - | 12KiB | 12KiB |
| | Traffic rate | 20.28tasks/s | $[650, 800]$tasks/s | 2000tasks/s |
| | Number of LB agents | 2 | 2 | 6 |
| | Total number of servers | 8 | 7 | 20 |
| | Server group 2 | 4 (1-CPU) | 3 (2-CPU) | 10 (2-CPU) |
| | Server group 1 | 4 (2-CPU) | 4 (4-CPU) | 10 (4-CPU) |
| | Episode duration | 60s | 60s | 60s |

Table 10: Four configurations with different application types.

| Application Type | Pure CPU | CPU Intensive | Balanced | IO Intensive |
| --- | --- | --- | --- | --- |
| Avg. CPU Time (s) | 1. | 0.75 | 0.5 | 0.25 |
| Avg. IO Time (s) | 0. | 0.25 | 0.5 | 0.75 |

# D   Hyperparameters

MARL-based load balancing methods are trained in both simulator, and moderate- and large-scale testbed setups for various amount of episodes. At the end of each episode, the RL models are trained and updated for 10 iterations. Given the total provisioned computational resource, the traffic rates of network traces for training are carefully selected so that the RL models can learn from sensitive cases where workloads should be carefully placed to avoid overloaded less powerful servers. The set of hyper-parameters are listed in Table 9.

# E   Results

## E.1   Inaccurate Server Weights

In real-world systems, not only error-prone configurations, but also different application profiles can lead to inaccurate server weight assignments. Using a similar setup where 2 cluster of 4 servers have the same IO processing speed but 2x different CPU processing speeds, different application profiles are compared to derive the actual server processing capacity differences. A 3-stage application whose queries follow CPU-IO-CPU processing stages is compared with a pure CPU application. Both CPU and IO processing time follow exponential distributions and the aggregated average FCT is 1s. The four different types of network applications are configured as in Table 10. As depicted in Fig. 2a, with different provisioned resource ratios for CPU (2x) and IO (1x) queues, to guarantee the optimal workload distribution fairness and make each server have the minised maximal remaining time to finish among all servers at all time, the weights to be assigned for servers with different provisioned capacities are stochastic and depend on different application profiles. Therefore, it is a sub-optimal solution for existing load balancing algorithms to statically configure server weights based on computational resources.

The setup in the paper for Table 3 is the following. There are 2 LB agents and 8 servers. 4 servers have 1 CPU worker-thread each while the other 4 servers have 2 CPU worker-threads each, to simulate

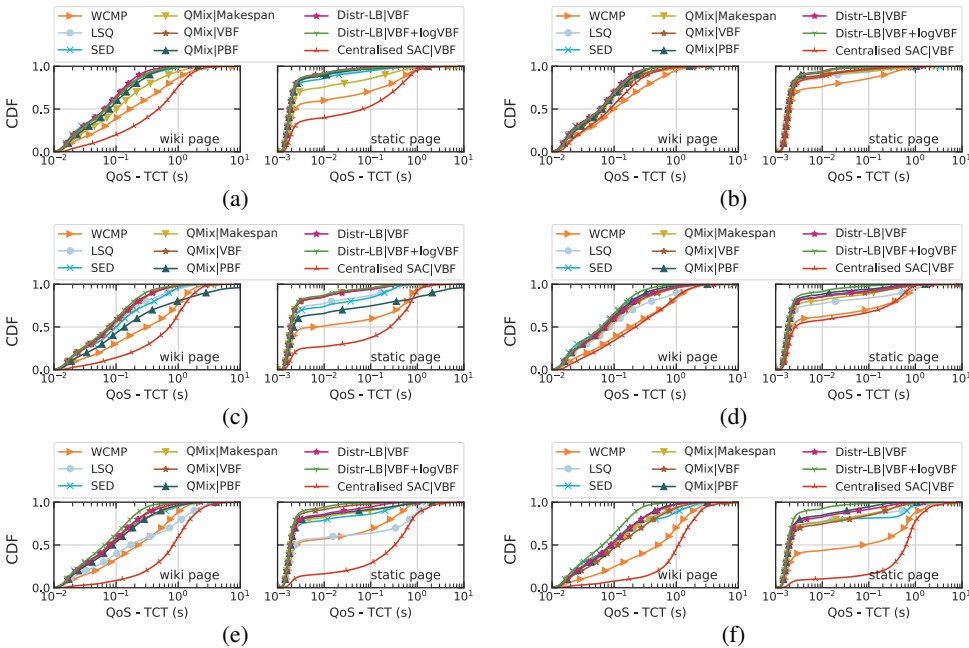

Figure 8: Experimental results with real-world network traces from different period of time during a day, which demonstrates the effectiveness of the proposed distributed RL framework with VBF as rewards.

the different server processing capacities. Three types of applications are compared. $100\%$-CPU application is a single stage application, whose expected time to process is $1$s in the CPU queue and $0$s in the IO queue. $75\%$-CPU+$25\%$-IO application is a two stage application, whose expected time to process is $0.75$s in the CPU queue and $0.25$s in the IO queue, simulating the CPU-intensive applications. $50\%$-CPU+$50\%$-IO application is a two stage application, whose expected time to process is $0.5$s in both the CPU and IO queue. The actual time to process for each task follows an exponential distribution. The traffic rate is normalised to consume on average $84.5\%$ resources.

## E.2    Ablation Results

Besides the experiments conducted in the paper, we further study the following aspects of the application of MARL on real-world network load balancing problems.

### E.2.1    Reward Engineering

To verify the effectiveness of the proposed potential function VBF, we compare it with a set of different reward functions, including makespan (MS), PBF, and coefficient of variation (CV). During our study based on real-world testbed, we found that, when using VBF as the reward, the convergence is fast at the beginning of the training process and the sample variance of average flow duration (as an estimation of the queuing and processing delay) on each server becomes close to zero. However, it does not necessarily mean that the load balancing policy is optimal and the NE is achieved. To capture the small variance which is close-to-zero, we also calculate the logarithm of VBF ($\log$VBF) as reward. And the combination of VBF + $\log$VBF is an empirical design aiming at faster convergence towards the NE policy. The complete comparison results are shown in Table 11 (average QoS) and in Table 12 (99th percentile QoS), where the proposed distributed MARL framework achieves the best performance for most cases. To provide a complete view of all comparison results besides the one shown in Fig. 4b, we show the CDF of task completion time under all test cases in Fig. 8. Accompanying the evaluation results of average QoS in large-scale testbed in Table 4, we also show in Table 13 the 99th percentile QoS in large-scale testbed.

Table 11: Complete results of *average* QoS (s) for comparison in moderate-scale real-world network setup (DC network and traffic).

| Method | | Period I (796.315 queries/s) | | Period II (687.447 queries/s) | | Period III (784.522 queries/s) | | Period IV (784.522 queries/s) | |
| --- | --- | --- | --- | --- | --- | --- | --- | --- | --- |
| | | Wiki | Static | Wiki | Static | Wiki | Static | Wiki | Static |
| WCMP | | 0.435 ± 0.083 | 0.171 ± 0.055 | 0.254 ± 0.087 | 0.073 ± 0.056 | 0.412 ± 0.101 | 0.134 ± 0.059 | 0.834 ± 0.323 | 0.492 ± 0.276 |
| LSQ | | 0.141 ± 0.073 | 0.023 ± 0.030 | 0.143 ± 0.040 | 0.023 ± 0.011 | 0.620 ± 0.442 | 0.339 ± 0.316 | 0.357 ± 0.373 | 0.173 ± 0.299 |
| SED | | 0.137 ± 0.076 | 0.020 ± 0.023 | 0.131 ± 0.067 | 0.027 ± 0.035 | 0.215 ± 0.210 | 0.051 ± 0.081 | 0.346 ± 0.496 | 0.169 ± 0.330 |
| RLB-SAC [40] | Jain | 0.137 ± 0.020 | 0.009 ± 0.006 | 0.125 ± 0.035 | 0.012 ± 0.008 | 0.193 ± 0.073 | 0.026 ± 0.022 | 0.204 ± 0.084 | 0.039 ± 0.047 |
| | G | 0.140 ± 0.053 | 0.015 ± 0.019 | 0.103 ± 0.022 | 0.010 ± 0.007 | 0.149 ± 0.049 | 0.015 ± 0.011 | 0.155 ± 0.052 | 0.011 ± 0.011 |
| QMix-LB | MS | 0.258 ± 0.174 | 0.071 ± 0.087 | 0.142 ± 0.073 | 0.030 ± 0.034 | 0.217 ± 0.157 | 0.048 ± 0.069 | 0.263 ± 0.202 | 0.073 ± 0.092 |
| | logMS | 0.167 ± 0.031 | 0.009 ± 0.004 | 0.132 ± 0.034 | 0.011 ± 0.008 | 0.844 ± 1.376 | 0.635 ± 1.249 | 0.278 ± 0.130 | 0.041 ± 0.038 |
| | VBF | 0.128 ± 0.052 | 0.014 ± 0.017 | 0.132 ± 0.075 | 0.016 ± 0.025 | 0.141 ± 0.025 | 0.008 ± 0.004 | 0.286 ± 0.162 | 0.068 ± 0.066 |
| | logVBF | **0.106 ± 0.011** | 0.007 ± 0.001 | 0.109 ± 0.032 | 0.011 ± 0.009 | 0.171 ± 0.043 | 0.022 ± 0.013 | 0.223 ± 0.045 | 0.026 ± 0.017 |
| | VBF+logVBF | 0.112 ± 0.005 | **0.005 ± 0.002** | 0.101 ± 0.010 | 0.005 ± 0.001 | 0.187 ± 0.090 | 0.024 ± 0.029 | 0.201 ± 0.080 | 0.021 ± 0.020 |
| | PBF | 0.142 ± 0.035 | 0.012 ± 0.006 | 0.099 ± 0.011 | **0.004 ± 0.001** | 0.211 ± 0.153 | 0.047 ± 0.078 | 0.181 ± 0.042 | 0.018 ± 0.009 |
| | CV | 0.407 ± 0.505 | 0.201 ± 0.340 | 0.113 ± 0.036 | 0.009 ± 0.008 | 0.203 ± 0.089 | 0.039 ± 0.037 | 0.219 ± 0.072 | 0.031 ± 0.017 |
| Centr-LB | VBF | 0.690 ± 0.211 | 0.284 ± 0.181 | 0.152 ± 0.041 | 0.016 ± 0.011 | 1.068 ± 0.386 | 0.570 ± 0.378 | 1.378 ± 0.377 | 0.867 ± 0.350 |
| | logVBF | 0.676 ± 0.231 | 0.265 ± 0.151 | 0.160 ± 0.023 | 0.013 ± 0.005 | 0.938 ± 0.200 | 0.446 ± 0.179 | 0.972 ± 0.288 | 0.495 ± 0.268 |
| | VBF+logVBF | 0.520 ± 0.034 | 0.167 ± 0.017 | 0.192 ± 0.040 | 0.019 ± 0.014 | 0.759 ± 0.254 | 0.306 ± 0.222 | 1.013 ± 0.168 | 0.520 ± 0.167 |
| Distr-LB (this paper) | VBF | **0.106 ± 0.013** | 0.007 ± 0.002 | 0.090 ± 0.016 | 0.007 ± 0.005 | 0.159 ± 0.054 | 0.017 ± 0.009 | 0.196 ± 0.091 | 0.032 ± 0.033 |
| | logVBF | 0.139 ± 0.021 | 0.011 ± 0.004 | 0.129 ± 0.032 | 0.012 ± 0.011 | 0.250 ± 0.156 | 0.057 ± 0.077 | 0.226 ± 0.059 | 0.038 ± 0.019 |
| | VBF+logVBF | 0.126 ± 0.038 | 0.009 ± 0.006 | 0.094 ± 0.023 | 0.006 ± 0.006 | **0.108 ± 0.022** | **0.004 ± 0.001** | **0.104 ± 0.013** | **0.006 ± 0.003** |
| | CV | 0.150 ± 0.040 | 0.011 ± 0.009 | 0.149 ± 0.060 | 0.026 ± 0.025 | 0.301 ± 0.146 | 0.066 ± 0.072 | 0.267 ± 0.156 | 0.051 ± 0.052 |

Table 12: Complete results of *99th percentile* QoS (s) for comparison in moderate-scale real-world network setup (DC network and traffic).

| Method | | Period I (796.315 queries/s) | | Period II (687.447 queries/s) | | Period III (784.522 queries/s) | | Period IV (784.522 queries/s) | |
| --- | --- | --- | --- | --- | --- | --- | --- | --- | --- |
| | | Wiki | Static | Wiki | Static | Wiki | Static | Wiki | Static |
| WCMP | | 5.801 ± 4.519 | 4.462 ± 3.867 | 4.019 ± 3.601 | 3.192 ± 3.543 | 3.239 ± 2.721 | 2.305 ± 2.700 | 8.066 ± 7.025 | 6.733 ± 5.329 |
| LSQ | | 0.722 ± 0.487 | 0.195 ± 0.314 | 0.814 ± 0.478 | 0.288 ± 0.259 | 1.846 ± 1.915 | 1.168 ± 1.575 | 1.257 ± 1.921 | 0.831 ± 2.002 |
| SED | | 0.706 ± 0.399 | 0.208 ± 0.246 | 0.697 ± 0.460 | 0.217 ± 0.291 | 0.726 ± 0.554 | 0.203 ± 0.261 | 0.909 ± 1.180 | 0.450 ± 1.112 |
| RLB-SAC [40] | Jain | 0.858 ± 0.240 | 0.159 ± 0.125 | 0.830 ± 0.358 | 0.227 ± 0.186 | 1.227 ± 0.489 | 0.354 ± 0.246 | 1.283 ± 0.594 | 0.408 ± 0.374 |
| | G | 0.945 ± 0.495 | 0.185 ± 0.214 | 0.682 ± 0.255 | 0.177 ± 0.162 | 1.003 ± 0.459 | 0.225 ± 0.176 | 0.973 ± 0.389 | 0.166 ± 0.156 |
| QMix-LB | MS | 1.469 ± 0.789 | 0.584 ± 0.547 | 1.095 ± 0.694 | 0.444 ± 0.423 | 1.182 ± 0.801 | 0.420 ± 0.483 | 1.447 ± 0.885 | 0.751 ± 0.772 |
| | logMS | 0.985 ± 0.264 | 0.117 ± 0.043 | 0.909 ± 0.388 | 0.172 ± 0.142 | 7.043 ± 12.237 | 6.427 ± 12.479 | 1.326 ± 0.584 | 0.371 ± 0.305 |
| | VBF | 0.732 ± 0.395 | 0.159 ± 0.239 | 0.665 ± 0.550 | 0.157 ± 0.278 | 0.744 ± 0.278 | 0.123 ± 0.093 | 1.028 ± 0.694 | 0.279 ± 0.365 |
| | logVBF | 0.682 ± 0.100 | 0.124 ± 0.019 | 0.772 ± 0.313 | 0.205 ± 0.159 | 1.174 ± 0.323 | 0.382 ± 0.183 | 1.426 ± 0.323 | 0.327 ± 0.153 |
| | VBF+logVBF | 0.664 ± 0.057 | **0.087 ± 0.056** | 0.611 ± 0.097 | 0.055 ± 0.027 | 1.171 ± 0.568 | 0.302 ± 0.293 | 1.206 ± 0.501 | 0.278 ± 0.239 |
| | PBF | 0.661 ± 0.193 | **0.087 ± 0.099** | 0.505 ± 0.119 | 0.048 ± 0.029 | 1.338 ± 0.759 | 0.205 ± 0.465 | 0.726 ± 0.433 | 0.128 ± 0.136 |
| | CV | 1.928 ± 2.228 | 1.281 ± 2.095 | 0.708 ± 0.405 | 0.131 ± 0.130 | 1.331 ± 0.593 | 0.481 ± 0.297 | 1.344 ± 0.329 | 0.451 ± 0.218 |
| Centr-LB | VBF | 3.101 ± 1.582 | 1.985 ± 1.790 | 0.903 ± 0.350 | 0.328 ± 0.353 | 4.409 ± 2.693 | 3.629 ± 3.219 | 6.649 ± 4.562 | 6.120 ± 4.721 |
| | logVBF | 2.715 ± 0.444 | 1.718 ± 0.547 | 1.016 ± 0.229 | 0.264 ± 0.092 | 3.247 ± 0.725 | 2.136 ± 0.832 | 4.286 ± 2.091 | 3.459 ± 2.323 |
| | VBF+logVBF | 2.459 ± 0.101 | 1.309 ± 0.063 | 1.243 ± 0.358 | 0.285 ± 0.189 | 2.796 ± 0.900 | 1.702 ± 1.287 | 3.466 ± 0.820 | 2.628 ± 1.142 |
| Distr-LB (this paper) | VBF | **0.651 ± 0.151** | 0.119 ± 0.072 | 0.571 ± 0.237 | 0.133 ± 0.136 | 1.039 ± 0.302 | 0.298 ± 0.125 | 1.187 ± 0.594 | 0.355 ± 0.318 |
| | logVBF | 0.923 ± 0.162 | 0.193 ± 0.086 | 0.933 ± 0.415 | 0.243 ± 0.302 | 1.491 ± 0.764 | 0.579 ± 0.531 | 1.481 ± 0.473 | 0.558 ± 0.286 |
| | VBF+logVBF | 0.745 ± 0.316 | 0.185 ± 0.152 | **0.385 ± 0.094** | **0.023 ± 0.003** | **0.595 ± 0.199** | **0.051 ± 0.030** | **0.563 ± 0.180** | **0.100 ± 0.073** |
| | CV | 0.865 ± 0.261 | 0.147 ± 0.121 | 1.109 ± 0.668 | 0.433 ± 0.431 | 1.730 ± 0.468 | 0.612 ± 0.420 | 1.383 ± 0.666 | 0.446 ± 0.345 |

### E.2.2 Communication Overhead of CTDE and Centralised RL

This section studies the communication overhead of CTDE RL scheme and analyses its impact on real-world distributed systems.

First, we discuss the communication overhead in data center networks in two-folds: throughput and latency.

1. **Thoughput:** Active signaling (*e.g.* periodically probing, or sharing messages) is an intrinsic way to observe and measure system states so that informed decisions can be made to improve performance [53, 54]. Higher communication frequency gives more relevant and timely observations yet there is a trade-off between communication frequency and additionally consumed bandwidth. Especially, in large distributed systems like data center networks, services are organized by multiple server clusters scattered all over the physical data center network in the era of cloud computing. Thus, management traffic among different nodes can cascade and plunder the bandwidth for data transmission in high-tier links. To demonstrate the trade-off between measurement quality and throughput overhead, we have conducted experiments to evaluate (i) the relevance of collected server utilization information to the actual server utilization information with root mean square error (RMSE) and Spearman's Correlation in our testbed on physical servers. When a controller VM periodically probes a server cluster via TCP sockets[8], as depicted in Fig. 9, the visibility over system states (relevance between measurements and ground truth) correlates with the probing frequency. Additional management traffic within a single service cluster —- behind one virtual IP (VIP) —- can exceed the 90th percentile of per-destination-rack flow rate (100kbps as depicted in Figure 8a in [45]) in Facebook data center in production.

As depicted in Fig. 10a, CTDE RL scheme requires agents to communicate and share their trajectories, which include the observed states and actions. This leads to linearly increasing replay buffer size with

---

[8]In the 69-byte control packet emitted by the server, the 24-byte payload consists of the server ID, CPU and memory usage, and the number of busy application threads.

Table 13: Comparison of 99th percentile QoS (s) in large-scale real-world network setup (DC network and traffic).

| Method | | Period I (2022.855 queries/s) | | Period II (2071.129 queries/s) | |
| --- | --- | --- | --- | --- | --- |
| | | Wiki | Static | Wiki | Static |
| WCMP | | $3.014 \pm 0.612$ | $2.152 \pm 0.907$ | $4.290 \pm 3.593$ | $3.300 \pm 3.308$ |
| LSQ | | $1.863 \pm 0.888$ | $0.843 \pm 0.773$ | $1.243 \pm 1.389$ | $0.675 \pm 1.223$ |
| SED | | $0.891 \pm 0.475$ | $0.208 \pm 0.251$ | $1.074 \pm 0.751$ | $0.592 \pm 0.650$ |
| RLB-SAC-G [40] | | $1.064 \pm 0.283$ | $0.210 \pm 0.132$ | $0.739 \pm 0.317$ | $0.186 \pm 0.214$ |
| **QMix-LB** | VBF | $1.104 \pm 0.481$ | $0.241 \pm 0.264$ | $1.223 \pm 1.169$ | $0.634 \pm 0.983$ |
| | PBF | $1.201 \pm 0.321$ | $0.196 \pm 0.112$ | $0.583 \pm 0.103$ | $0.071 \pm 0.050$ |
| **Distr-LB** | VBF | $1.350 \pm 0.311$ | $0.263 \pm 0.139$ | $1.180 \pm 0.702$ | $0.448 \pm 0.371$ |
| (this paper) | VBF+logVBF | **$0.890 \pm 0.250$** | **$0.103 \pm 0.064$** | **$0.531 \pm 0.149$** | **$0.057 \pm 0.039$** |

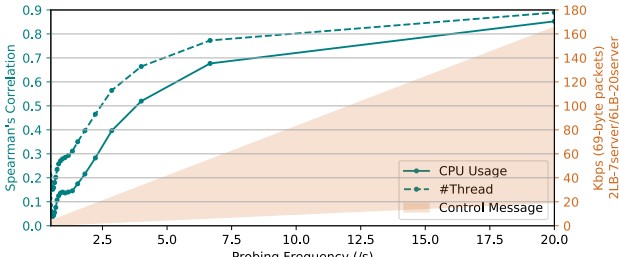

Figure 9: Correlation (Spearman) increases when the probing frequency grows, yet, so do additional control messages.

the growth of number of episodes. The replay buffer size also grows with the number of agents which makes CTDE RL scheme not a scalable mechanism. Transmitting and synchronising replay buffer among agents incur additional communication overhead in the networking system, reducing the throughput for data transmission channel – which can break full-bisection bandwidth (an important throughput related performance metric) in data center networks [55] – thus decreasing the QoS.

2. **Latency:** Using the same network topology as the moderate-scale real-world testbed, when a single controller VM periodically transmit different amount of bytes via TCP sockets towards the agents, the latency overhead increases with the number of servers, which diminishes the QoS, as depicted in Fig. 10b. It is measured for per-packet round trip time (RTT) between two directly connected network nodes. While normal RTT is $0.099ms \pm 0.014ms$ in such setup, with additional communication overhead, RTT can grow more than 10x. This is not considered as low additional latency, especially not in high performance networking systems. In elastic and cloud computing context and real-world setups, load balancers can be deployed in different racks [56]. There can be multiple hops between two nodes and one connection consists of tens of hundreds of packets, which can lead to cascaded high latency.

Based on the analysis of Fig. 9, we can see that delayed measurement and communication can cause degraded system state observation. To further demonstrate the performance of the passive feature collection mechanism which incurs absolutely zero communication overhead, an additional experiment is conducted to compare the feature collection latency. The latency overhead of passive feature collection process in our paper using POSIX shared memory is compared with different active probing techniques. The idle communication latency is compared using both KVM and Docker containers between two hosts either deployed on the same machine (local) or on two neighbor machines

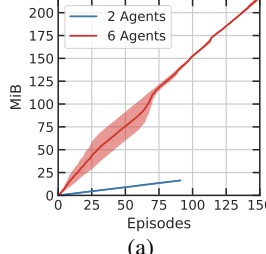
(a)

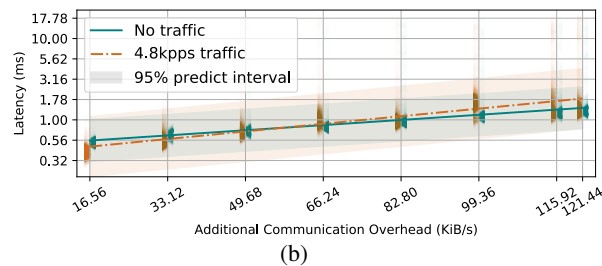
(b)

Figure 10: Communication overhead for CTDE (a) grows linearly during training and (b) can have negative effects on the packet transmission latency of the whole networking system.

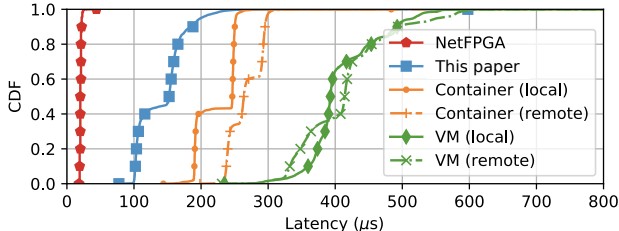

Figure 11: Feature collection latency comparison against active probing techniques.

Table 14: Comparison of average job completion time (s) of static pages under different traffic rates using large-scale real-world setup.

| Method | | Traffic Rate (queries/s) | | | | | | | | |
|---|---|---|---|---|---|---|---|---|---|---|
| | | 731.534 | 1097.3 | 1463.067 | 1828.834 | 2194.601 | 2377.484 | 2560.368 | 2743.251 | 2926.135 |
| LSQ | | 0.048 ±0.002 | 0.055 ±0.003 | 0.059 ±0.003 | 0.069 ±0.008 | 0.131 ±0.070 | 0.643 ±0.325 | 1.910 ±0.269 | 2.873 ±0.215 | 3.545 ±0.146 |
| SED | | 0.054 ±0.001 | 0.061 ±0.004 | 0.068 ±0.004 | 0.080 ±0.004 | 0.117 ±0.025 | 0.660 ±0.396 | 1.718 ±0.366 | 2.767 ±0.207 | 3.482 ±0.189 |
| **Distr-LB** (this paper) | VBF | 0.047 ±0.001 | 0.054 ±0.003 | 0.059 ±0.005 | **0.066** **±0.007** | 0.105 ±0.035 | **0.266** **±0.139** | 1.465 ±0.115 | 2.047 ±0.145 | 2.704 ±0.108 |
| | VBF+logVBF | 0.047 ±0.001 | 0.054 ±0.004 | 0.059 ±0.004 | 0.069 ±0.008 | **0.084** **±0.009** | 0.413 ±0.249 | **1.183** **±0.063** | **1.838** **±0.083** | **2.513** **±0.105** |

(remote). To compare with the shortest latency possible of an hardware-based SDN controller directly connected to the agent, a loopback test is conducted using a NetFPGA [57] connected to the machine via both Ethernet and PCIe. We parse features stored in the local shared memory with a simple Python script without generating control messages. As depicted in Fig. 11, its median processing latency outperforms typical VM- and container-based VNF probing mechanisms [58–61] by more than $94.18\mu s$.

To evaluate the performance of the proposed algorithm within the operational range of traffic rates, we conducted the scaling experiment using 6 LB agents and 20 servers in the real-world testbed with traffic rates that range from low to high. As shown in Table 14 and 15, similar to the QoS (99-th percentile of the task completion time) evaluation in Table 5 and 6. Low traffic rates do not saturate server processing capacities and the servers are not stressed. Therefore, all servers are able to handle all the requests without accumulating jobs in the queue regardless of the differences of their processing capacities. However, under heavy traffic rates, LSQ still distribute workloads so as to maintain the same queue lengths on servers with different processing speeds, which leads to degraded average task completion time. SED assigns more jobs in proportional to the number of CPUs deployed for each server, achieving slightly better performance than LSQ in terms of the average task completion time. The proposed Distr-LB outperforms both LSQ and SED especially under heavy traffic rates, thus when servers undergo heavy resource utilisation. Since the server processing speed for different applications is not necessarily proportional to the number of CPU–as we have discussed over Fig. 2a in Sec. 2, Distr-LB is able to learn the appropriate ratio of workload distribution for servers with different capacities.

Table 15: Comparison of average job completion time (s) of static pages under different traffic rates using large-scale real-world setup.

| Method | | Traffic Rate (queries/s) | | | | | | | | |
|---|---|---|---|---|---|---|---|---|---|---|
| | | 731.534 | 1097.3 | 1463.067 | 1828.834 | 2194.601 | 2377.484 | 2560.368 | 2743.251 | 2926.135 |
| LSQ | | 0.004 ±0.001 | 0.004 ±0.000 | 0.003 ±0.000 | 0.004 ±0.000 | 0.018 ±0.023 | 0.252 ±0.234 | 1.455 ±0.258 | 2.426 ±0.207 | 3.080 ±0.136 |
| SED | | 0.003 ±0.000 | 0.004 ±0.001 | 0.004 ±0.000 | 0.004 ±0.000 | 0.006 ±0.003 | 0.284 ±0.308 | 1.283 ±0.374 | 2.322 ±0.226 | 3.041 ±0.188 |
| **Distr-LB** (this paper) | VBF | 0.004 ±0.000 | 0.004 ±0.000 | 0.004 ±0.000 | 0.004 ±0.000 | **0.005** **±0.001** | **0.055** **±0.070** | 1.039 ±0.144 | 1.617 ±0.135 | 2.277 ±0.096 |
| | VBF+logVBF | 0.004 ±0.000 | 0.004 ±0.000 | 0.004 ±0.000 | 0.004 ±0.000 | 0.006 ±0.004 | 0.116 ±0.114 | **0.750** **±0.063** | **1.413** **±0.083** | **2.076** **±0.096** |

Table 16: Comparison of QoS (mean, 95th-percentile, and 99th-percentile task completion time in $s$) when server processing capacity changes over time.

| | | Wiki | | | Static | | |
|---|---|---|---|---|---|---|---|
| | | Mean | $95th$-percentile | $99th$-percentile | Mean | $95th$-percentile | $99th$-percentile |
| WCMP | | $1.792 \pm 0.393$ | $7.534 \pm 1.817$ | $2.366 \pm 1.685$ | $1.512 \pm 0.385$ | $6.571 \pm 1.996$ | $1.084 \pm 1.842$ |
| LSQ | | $0.453 \pm 0.178$ | $1.958 \pm 0.827$ | $3.482 \pm 1.257$ | $0.202 \pm 0.130$ | $0.975 \pm 0.617$ | $1.801 \pm 1.064$ |
| SED | | $0.340 \pm 0.268$ | $1.225 \pm 0.812$ | $30.600 \pm 6.718$ | $0.130 \pm 0.206$ | $0.519 \pm 0.571$ | $29.893 \pm 7.042$ |
| **QMix-LB** | MS | $0.373 \pm 0.177$ | $1.621 \pm 0.830$ | $4.046 \pm 6.632$ | $0.144 \pm 0.112$ | $0.663 \pm 0.523$ | $2.655 \pm 6.899$ |
| | PBF | $0.368 \pm 0.375$ | $1.529 \pm 1.581$ | $2.436 \pm 1.468$ | $0.159 \pm 0.338$ | $0.733 \pm 1.437$ | $0.974 \pm 1.204$ |
| | VBF | $0.282 \pm 0.166$ | $1.186 \pm 0.799$ | $3.187 \pm 1.479$ | $0.081 \pm 0.104$ | $0.395 \pm 0.518$ | $1.654 \pm 1.181$ |
| | VBF+logVBF | $0.533 \pm 0.179$ | $2.525 \pm 0.913$ | $4.864 \pm 1.635$ | $0.266 \pm 0.129$ | $1.409 \pm 0.680$ | $3.374 \pm 1.626$ |
| **Distr-LB** | VBF | $0.262 \pm 0.100$ | $1.086 \pm 0.454$ | $2.190 \pm 0.792$ | $0.057 \pm 0.044$ | $0.305 \pm 0.234$ | $0.683 \pm 0.510$ |
| (this paper) | VBF+logVBF | $\mathbf{0.221 \pm 0.112}$ | $\mathbf{0.895 \pm 0.530}$ | $\mathbf{1.903 \pm 0.976}$ | $\mathbf{0.039 \pm 0.057}$ | $\mathbf{0.197 \pm 0.284}$ | $\mathbf{0.480 \pm 0.650}$ |

### E.2.3 MARL Robustness

With the rise of elastic and server-less computing, where tenants in data center can share physical resources (*e.g.* CPU, disk, memory), servers can have different processing capacities, which may also change over time dynamically —- because of *e.g.* updated server configuration (upgrading an Amazon EC2 `a1.xlarge` instance to `a1.4xlarge`) or resource contention (co-located workloads) [62]. According to [14], there are 32% of server clusters in data center that update more than 10 times per minute based on the measurements collected over 432 minutes up time in a month. 3% of clusters have more than 50 updates perf minute. Therefore, dynamic changes prevail in real-world data center networks.

Therefore, this section studies the robustness of the proposed distributed RL-based LB framework to react to dynamic changes in server processing speeds, *e.g.* when server VMs are migrated to a new physical architecture. Using the same moderate-scale real-world testbed with 2 LB agents, additional CPU-bound workloads are applied on the 4-CPU server group starting from 25s. As depicted in Fig. 12a, under heavy Wikipedia traffic, MARL-based LB agents adapt server weights over time and achieves better performance than heuristic LB algorithms – finishing the same amount of workloads faster, maintaining lower amount of acive number of threads, even when server processing capacity is reduced. As depicted in Fig. 12b, over multiple runs (10 runs for each LB algorithm), RL-based LB algorithms effectively achieves lower task completion time in dynamic environments. They help avoid human intervention and make the LB agents autonomously adapt to the changes in the system. Table 16 lists the performance of all LB algortihms in terms of the QoS (measured as the average and 95th-percentile task completion time).

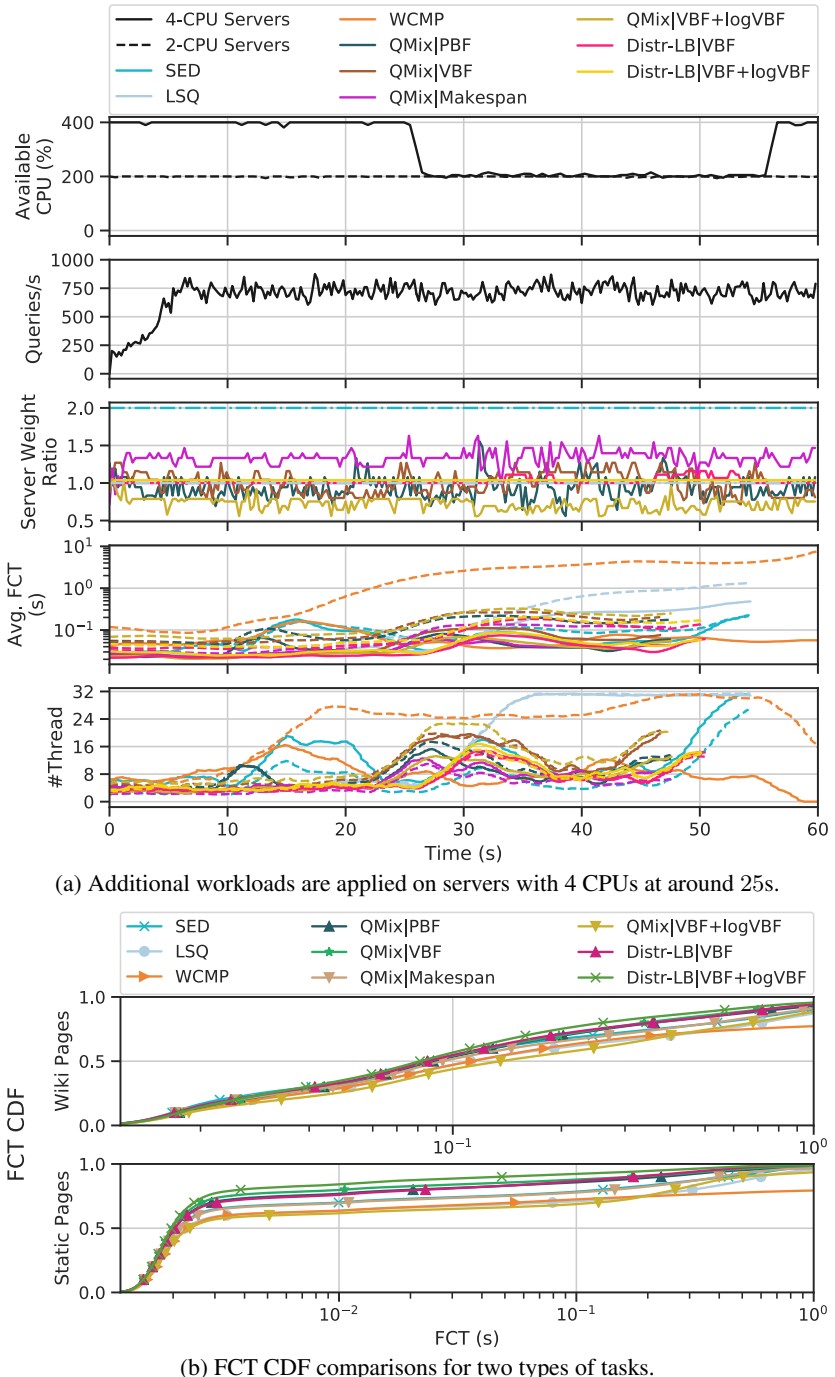

(a) Additional workloads are applied on servers with 4 CPUs at around 25s.

(b) FCT CDF comparisons for two types of tasks.

Figure 12: Load balancing performance comparison in dynamic environments.