# OpenReview forum: "Learning Distributed and Fair Policies for Network Load Balancing as Markov Potential Game"
_NeurIPS.cc/2022/Conference — NeurIPS 2022 Accept_

### Official Review · Reviewer_WwLw · 2022-07-09

**Rating:** 5
**Confidence:** 4
**Soundness:** 2 fair
**Presentation:** 3 good
**Contribution:** 2 fair

**Summary:**

The paper explores the task of multi-agent network load balancing via formulation as a Markov potential game, using workload distribution fairness as a potential function. A MARL algorithm is proposed based on this formulation and provides for fully-decentralized learning. The paper further develops an event-based simulator which, along with a real-world network setup, is used to evaluate the proposed algorithm against several MARL baselines.

**Questions:**

- Is there any reason why the work cannot compare against more recent CTDE schemes?
- Comments on other concerns?


**Strengths And Weaknesses:**

Strengths:
+ Rigorous formulation of network load balancing as MPG with proofs that appear sound.
+ Generally interesting and well-motivated application for MARL with promising potential

Weaknesses:
- Concern regarding representativeness of baselines used for evaluation
- Practical benefits in terms of communication overhead & training time could be more strongly motivated

Detailed Comments:

Overall, the paper was interesting to read and the problem itself is well motivated. Formulation of the problem as an MPG appears sound and offers a variety of important insights with promising applications. There are, however, some concerns regarding evaluation fairness and practical benefits.

The baselines used for evaluation do not seem to accurately represent the state-of-the-art in CTDE. In particular, there have been a variety of recent works that explore more efficient strategies (e.g., [1-3]) and consistently outperform QMix with relatively low inter-agent communication. Although the proposed work appears effective as a fully-decentralized approach, it is unclear how well it would perform against more competitive CTDE baselines. Comparison against these more recent works would greatly improve the strength of evaluation.

Benefits in terms of reduced communication overhead could also be more strongly motivated. Presumably, communication between agents could be done over purpose-built inter-LB links, thus avoiding QoS degradation due to contention on links between LBs and servers. Even without inter-LB links, the increase in latency demonstrated in Appendix E.2.2 appears relatively low.

Robustness against dynamic changes in network setup are discussed to some degree, but it’s unclear how significant this issue is in a real-world environment. Even in a large-scale setup, the number of LBs/servers is likely to remain fairly constant at the timescales considered in this work (i.e., minutes). Given this, it seems that the paper should at least discuss trade-offs with a longer training time, which could impact the relative benefits of various approaches.

Some confusion in notation:
-	Algorithm 2, L8 should be t = 1,…,H (for horizon)?
-	L100, [M] denotes the set of LBs?

Minor notes:
-	Some abbreviations are not defined, e.g., “NE” on L73
-	Superscript notation in Eq 6 is not defined until much later (L166), which hindered understanding in an initial read.

[1] S. Zhang et al, “Efficient Communication in Multi-Agent Reinforcement Learning via Variance Based Control”, NeurIPS 2019.
[2] Z. Ding et al, “Learning Individually Inferred Communication for Multi-Agent Cooperation”, NeurIPS 2020.
[3] T. Wang et al, “Learning Nearly Decomposable Value Functions Via Communication Minimization”, ICLR 2020.

---

> ### Author Response · Authors · 2022-08-02
> **First response to Reviewer WwLw (1/2)**
>
> Dear Reviewer WwLw,
>
> Thank you for your time to review our paper and we appreciate your thorough and constructive comments.
>
> > Representativeness of baselines used for evaluation: The baselines used for evaluation do not seem to accurately represent the state-of-the-art in CTDE e.g., [1-3].
> >
>
> ***Response:*** We appreciate the reviewer for pointing us to very interesting CTDE works [1-3]. These works target cooperative setups and propose to exploit (minimized) communications among agents during the “distributed execution” phase to improve performance and reduce communication overhead. However, in deployed load balancing systems in real-world data center environments — or distributed execution phase, no direct communication happens between agents [6-10], because load balancers are either fully distributed (self-managed) or directly managed by a centralized controller to reduce management and operational overhead in data center networks. Even if a direct inter-load-balancer channel was to be established, in elastic data centers, two load balancers may be deployed in different racks [9], which can cause (i) delayed signal (>1ms) because of multi-hop distance while inter-task arrival time can be < 200 microseconds, and (ii) additional bandwidth consumption in high-tier links in physical data centers (discussed later). Besides, [2] defines the field of view for each agent, which does not apply in data center network load balancing setups, and [3] introduces another parameter (message drop rate) to tune for achieving optimized performance. Presumably, if we were to implement these CTDE algorithms, more sensitivity analysis needs to be conducted to study the impact of *e.g.*, frequency of communication among agents, and types of information communicated, which we consider will hinder the clarity of the paper. However, this comment can be really valuable for the MARL application in mobile networks.
>
> Following the suggestion of reviewer Y8vv, we have conducted additional experiments using RLB-SAC (NeurIPS 2021) — the latest RL-based solution for network load balancing. We respectfully refer the reviewer to our answer to reviewer Y8vv as well as our updated results in the revised paper, to check the results, which demonstrate that our proposed method has superior performance.
>
> > Benefits in terms of reduced communication overhead could also be more strongly motivated. Presumably, communication between agents could be done over purpose-built inter-LB links, thus avoiding QoS degradation due to contention on links between LBs and servers. Even without inter-LB links, the increase in latency demonstrated in Appendix E.2.2 appears relatively low.
> >
>
> ***Response:*** This is a very valuable remark. We have provided more motivation for reduced communication overhead and argued that the additional latency is not relatively low in our answer to **Common Concern 2** in our **Meta Response** above.
>
> (to be continued)
>
> ---
>
> Reference:
>
> [1] S. Zhang et al, “Efficient Communication in Multi-Agent Reinforcement Learning via Variance Based Control”, NeurIPS 2019.
>
> [2] Z. Ding et al, “Learning Individually Inferred Communication for Multi-Agent Cooperation”, NeurIPS 2020.
>
> [3] T. Wang et al, “Learning Nearly Decomposable Value Functions Via Communication Minimization”, ICLR 2020.
>
> [6] Eisenbud, Daniel E., et al. "Maglev: A fast and reliable software network load balancer." *13th USENIX NSDI 16*. 2016.
>
> [7] Barbette, Tom, et al. "A High-Speed Load-Balancer Design with Guaranteed Per-Connection-Consistency." *17th USENIX NSDI*. 2020.
>
> [8] Olteanu, Vladimir, et al. "Stateless datacenter load-balancing with beamer." *15th USENIX NSDI*. 2018.
>
> [9] Gandhi, Rohan, et al. "Duet: Cloud scale load balancing with hardware and software." *ACM SIGCOMM*  2014
>
> [10] Desmouceaux, Yoann, et al. "6lb: Scalable and application-aware load balancing with segment routing." *IEEE/ACM Transactions on Networking* 2018

---

> > ### Author Response · Authors · 2022-08-02
> > **First response to Reviewer WwLw (2/2)**
> >
> > (cont'd)
> >
> > > Robustness against dynamic changes in network setup is discussed to some degree, but it’s unclear how significant this issue is in a real-world environment. Even in a large-scale setup, the number of LBs/servers is likely to remain fairly constant at the timescales considered in this work (i.e., minutes). Given this, it seems that the paper should at least discuss trade-offs with a longer training time, which could impact the relative benefits of various approaches.
> > >
> >
> > ***Response:*** With the rise of elastic and server-less computing, where tenants in the data center can share physical resources (*e.g.,* CPU, disk, memory), servers can have different processing capacities, which may also change over time dynamically — because of e.g., updated server configuration (upgrading an Amazon EC2 a1.xlarge instance to a1.4xlarge) or resource contention (co-located workloads) [4]. According to [5], there are 32% of server clusters in a data center that update more than 10 times per minute based on the measurements collected over 432 minutes up time in a month. 3% of clusters have more than 50 updates per minute. Therefore, dynamic changes prevail in real-world data center networks. The episode length — because of the variance in traffic rate and load balancing decisions — varies from 45s-75s, which captures a good representation of various scenarios, meanwhile allowing for efficiently collecting playback trajectories. We have carefully tuned the training time in our paper so that different load balancing methods converged (as depicted in Fig. 4 in the main paper) and no significant improvement happens with longer training time.
> >
> > ---
> >
> > Given the clarifications and additional experimental results detailed above, we kindly ask the reviewer to re-evaluate our paper and consider recommending it for acceptance.
> >
> > We thank you in advance for your reply. We hope our responses and edits help convince you of the merits of this paper. Please do not hesitate in contacting us during the rolling discussion and let us know if there is anything else that we can revise.
> >
> > ---
> >
> > Reference:
> >
> > [4] Guo, Jing, et al. "Who limits the resource efficiency of my datacenter: An analysis of alibaba datacenter traces." IEEE/ACM IWQoS. 2019.
> >
> > [5] Miao, Rui, et al. "Silkroad: Making stateful layer-4 load balancing fast and cheap using switching asics." *SIGCOMM*. 2017.

---

> ### Author Response · Authors · 2022-08-08
> **Reminder: updated feedback**
>
> Dear Reviewer WwLw,
>
> Thank you again for your review.
>
> We are sorry that we have not heard back from you since we submitted our rebuttals. The discussion period with the authors is coming to an end and we would not like to miss your response due to time difference. Would you mind please giving us an update on your new interpretation of our work given the revision and the information from our first response?
>
> What would it take for us to get a good recommendation from you at this point?
>
> Thank you very much in advance for your time and help!
>
> Sincerely,
>
> Paper10297 Authors

---

> > ### Comment · Reviewer_WwLw · 2022-08-09
> > **Thank you for clarification**
> >
> > Dear Authors, Thank you for responding to my questions. I appreciate your detailed explanation and find the clarification helpful. The additional results also help to alleviate my concerns. I will increase my rating to Boarderline Accept.

---

### Official Review · Reviewer_yzYr · 2022-07-17

**Rating:** 6
**Confidence:** 3
**Soundness:** 2 fair
**Presentation:** 4 excellent
**Contribution:** 3 good

**Summary:**

This paper proposes a distributed Multi-agent Reinforcement Learning based approach for load balancing at the network layer formulated as a Markov Potential game.

Current network load balancers have limited observability over the workloads and servers performance and are prone to misconfiguration due to heterogeneity and elasticity. Centralized approaches (CTDE) incur an additional overhead from centralized communication. This work addresses this by using a local variance-based fairness function in each load balancer which, when maximized, can minimize the potential function of the Markov potential game. This approximates the Nash equilibrium of the game.


**Questions:**

* This work addresses load balancing in a single data center but I think it can easily be extended to a multi-DC setup by capitalizing on the low-observability approximation. Have you considered this?
* Have you considered including some offline learning as the intial state? System logs from data centers usually provide good information about request patterns and existing data center hardware.
* What is the rationale behind using 2/3 agents in Fig 2b?
* Did you consider other kinds of traffic other than wiki (database heavy) and static? (video for example)
* Are the numbers in Tables 1/2 mean or median?
* Nitpick: Page 3, line 100 should say [M] denotes the set of LB agents {1,...,M} instead of set of servers
* Nitpick: Specs for the real world setup would be helpful to compare to existing DCs


**Limitations:**

* The paper doesn't seem to address elastic setups even though their motivation included both heterogeneous and elastic infrastructures.
* Simulator not as complex as real-world (addressed in paper). Still allows to test parts of the system without stochastic network parameters. These could be synthetically injected though.
* Need for low communication overhead in DC is not motivated strongly. Centralized methods (QMix for example) still show comparable performance in some application setups.


**Strengths And Weaknesses:**

## Strengths
* Significant gains by using proposed design over current in-production load balancing algorithms
* Strong theoretical foundation of formulating load balancing as a multi-agent RL-based Markov potential game
* Well written paper that puts the pieces of the design in an easy to understand order

## Weaknesses
* DCs typically have high bandwidth for internal communication. The paper states that centralized communication leads to heavy overhead which is not convincing in the main paper. The evaluation section mentions it in passing as being evaluated in the appendix but I feel it would be helpful to show in the main paper.
* Not sure if I agree with "large-scale" DC networks having only 20 servers. The largest of data centers have thousands of servers and load balancers. This makes the real-world setup slightly less impressive.
* Fault tolerance not evaluated in the paper (in terms of failed requests leading to incorrect job completion estimates for next time period, network partitions etc.)

---

> ### Author Response · Authors · 2022-08-02
> **First response to Reviewer yzYr (1/2)**
>
> Dear Reviewer yzYr,
>
> Thank you for taking the time to review our paper and we appreciate your insightful comments.
>
> > DCs typically have high bandwidth for internal communication. The paper states that centralized communication leads to heavy overhead which is not convincing in the main paper. The evaluation section mentions it in passing as being evaluated in the appendix but I feel it would be helpful to show in the main paper.
> >
>
> > Need for low communication overhead in DC is not motivated strongly. Centralized methods (QMix for example) still show comparable performance in some application setups.
> >
>
> ***Response:*** This is a very valuable remark. We respectfully refer the reviewer to our answer to **Common Concern 2** in the **Meta Response** to all reviewers.
>
> > Not sure if I agree with "large-scale" DC networks having only 20 servers. The largest of data centers have thousands of servers and load balancers. This makes the real-world setup slightly less impressive.
> >
>
> > Nitpick: Specs for the real world setup would be helpful to compare to existing DCs
> >
>
> ***Response:*** This is a reasonable comment. We have addressed this comment in our answer to **Common Concern 1** in our **Meta Response** above. We have also added a brief survey as specs for the real-world setup in DCs.
>
> > Fault tolerance not evaluated in the paper (in terms of failed requests leading to incorrect job completion estimates for the next time period, network partitions, etc.)
> >
>
> ***Response:*** This is a valuable remark. We agree that fault tolerance is worth proper investigation. However, to thoroughly system robustness and fault tolerance requires additional evaluation and analysis (which can be a standalone topic to study as in [1]), which, we estimate, would hinder the clarity of this paper. Methods like robust RL would be considered or compared in this setting. In this paper, we have conducted a system robustness evaluation, which will be described in the answer to the following question.
>
> In terms of collecting measurements of job completion time which might be corrupted by failed requests, we refer the reviewer to our contribution described in Sec. C2.4. in the revised supplementary material. We use reservoir sampling to gather task completion time (and task duration). The basic idea is to have a fixed-sized (size k = 64 in the paper) buffer for each feature for each server behind a given service (or virtual IP). Whenever a new data point is received at time $t$ (*e.g.*, when a request finished with a TCP FIN packet), the task completion time — together with the timestamp $t$ — is inserted in a random bucket in the buffer, replacing the previously stored value. Therefore, given a Poisson stream of events with an arrival rate $\lambda$, reservoir sampling gathers an exponentially distributed number of samples over a time window — the expectation of the number of samples that are preserved in the buffer after $n$ steps is $E = \lambda(\frac{k-1}{k})^{\lambda n}$. At every time step, the reservoir buffer is summarized by calculating its mean, standard deviation, discounted average, percentiles, etc. as input features. Thus, a couple of failed requests will have little impact on the overall measurement representation. However, this reservoir sampling mechanism still allows for collecting timeouts in the buffer and detecting issues if most requests fail because of overloaded servers.
>
> > The paper doesn't seem to address elastic setups even though their motivation included both heterogeneous and elastic infrastructures.
> >
>
> ***Response:*** We agree that it is important to evaluate elastic setups, which also, in part, motivated us to add a detailed discussion about elastic DC setups in our answer to **Common Concern 1** in our **Meta Response**. To study the system in dynamic environments in the context of elastic computing, we have presented experimental evaluations in Sec. E.2.3 in the supplementary material, where server processing capacities change over time. We consider evaluating more elastic setups (*e.g.*, adding or removing servers with action masking) for future work.
>
> (to be continued)
>
> ---
>
> Reference:
>
> [1] Ghaznavi, Milad, et al. "Fault tolerant service function chaining." *SIGCOMM*. 2020.

---

> > ### Author Response · Authors · 2022-08-02
> > **First response to Reviewer yzYr (2/2)**
> >
> > (cont'd)
> >
> > > Other future work directions:
> > >
> > >
> > > > This work addresses load balancing in a single data center but I think it can easily be extended to a multi-DC setup by capitalizing on the low-observability approximation.
> > > >
> > >
> > > > Have you considered including some offline learning as the initial state?
> > > >
> > >
> > > > Simulator not as complex as real-world (addressed in paper). Still allows testing parts of the system without stochastic network parameters.
> > > >
> >
> > ***Response:*** We appreciate the insightful comments and inspiring suggestions.
> >
> > - Multi-DC setup or inter-DC use cases can be interesting, especially for traffic optimization and balancing traffic among PoP (point of presence) data centers.
> > - Offline training has been used by AWS EC2 to learn from history data *e.g.*, diurnal traffic patterns. We believe that it is also possible to conduct a similar pre-training process to learn different types of hardware. However, there are also recent works that challenge the assumption of consistent patterns over time [2]. And there are more challenges to tackle including the dynamically changing environment (frequently and elastically updated server clusters [3].
> > - We agree that more sensitivity analysis (e.g., by varying CPU/IO, traffic characteristics) will be interesting. However, to thoroughly evaluate what happens with extensive parameter tuning in terms of testbed configuration requires additional modeling, evaluation, and analysis, which, we estimate, would hinder the clarity of this paper.
> >
> > We have already open-sourced both our simulator and testbed in case more researchers would like to join forces for future work concerning these points.
> >
> > > What is the rationale behind using 2/3 agents in Fig 2b?
> > >
> >
> > ***Response:*** In Fig. 2b, we wanted to demonstrate with a simplistic case, that in real-world data centers where multiple load balancers distribute workloads among a server cluster, partial observation will degrade the performance of classical methods. Therefore, we proposed this MARL framework with a carefully designed potential function to handle the partial observation issue and achieved improved performance when compared with SOTA load balancer methods.
> >
> > ---
> >
> > Given the clarifications, we kindly ask the reviewer to re-evaluate our paper and consider recommending it for acceptance. We hope to have answered your questions sufficiently and helped you see the merits of this paper in a new light. We look forward to your feedback.
> >
> > ---
> >
> > Reference:
> >
> > [2] Jajoo, Akshay, et al. "A case for task sampling-based learning for cluster job scheduling." *19th USENIX NSDI 22*. 2022.
> >
> > [3] Miao, Rui, et al. "Silkroad: Making stateful layer-4 load balancing fast and cheap using switching asics." *SIGCOMM*. 2017.

---

> ### Author Response · Authors · 2022-08-08
> **Reminder: updated feedback**
>
> Dear Reviewer yzYr:
>
> Thanks again for your review with constructive comments. We hope our answers could increase your confidence. As the discussion period is close to the end and we have not yet heard back from you, we would be glad to see if our rebuttal response has addressed your questions/concerns.
>
> We are more than happy to discuss if you have any further questions, please kindly let us know. Thank you for your time and help!
>
> Sincerely,
>
> Paper10297 Authors

---

### Official Review · Reviewer_cVpL · 2022-07-18

**Rating:** 5
**Confidence:** 4
**Soundness:** 2 fair
**Presentation:** 3 good
**Contribution:** 2 fair

**Summary:**

This paper considers the load balancing problem in a network of multiple heterogenous servers, and multiple load balancers.  The authors formulate the problem as a multi agent reinforcement learning problem, and specifically consider a Markova potential game.

The settings is that of multiple load balancers, each responsible for sending jobs to a set of servers.  There might be overlaps in the set of servers the various load balancers serve, and the load balancers thus have partial observability of the system state.  Using the cumulative total fairness as the potential function, where fairness is defined as either variance fairness or product fairness, the authors show that the job allocation game where the objective is to minimize the makespan while maximizing the variance fairness or product fairness, is a Markov potential game.

**Questions:**

Are all servers allocated jobs by all LBers?
What do the results look like with partial overlaps? This seems to be a harder problem.

**Limitations:**

The experimentation doesn’t include comparison with classical methods such as LSQ.  It would be interesting to see how a distributed, blind, greedy LSQ compares to the distributed MARL method proposed here, especially since the computation costs are so vastly different.

**Strengths And Weaknesses:**

A network with multiple load balancers managing load to multiple and overlapping servers is a complex problem.  The interactions between the load balancers is such that a closed form solution to the balancing problem is not evident.  This approach of setting a potential game within a RL environment is interesting and seems novel.

The authors propose a distributed load balancing method where each agent independently learns a policy, through policy gradient methods. The reward function is set to be per-LB variance or product fairness.  The authors show that maximizing for these local fairness metrics is sufficient tor minimizing makespan, a global metric.
The exact model, with respect to overlap of servers among the load balancers, is not clear.  Are all servers allocated jobs by all LBers? What do the results look like with partial overlaps? This seems to be a harder problem.

---

> ### Author Response · Authors · 2022-08-02
> **First response to Reviewer cVpL**
>
> Dear Reviewer cVpL
>
> Thank you for taking the time to review our paper and we appreciate your insightful comments.
>
> > Are all servers allocated jobs by all LBers? What do the results look like with partial overlaps? This seems to be a harder problem.
> >
>
> ***Response:*** Yes, all servers are allocated jobs by all load balancers. Because in the context of data center networks, the aim is to achieve as high throughput as possible with multiple load balancers, so that the uplink bandwidth matches the server-facing bandwidth. For instance, given a $12$ $1$Gbps server cluster, we need a $3$ $4$Gbps load balancer so that the load balancers will not become a bottleneck in the network even when all servers go full throttle. Therefore, in all SOTA load balancing mechanisms in data centers, load balancers are connected to all servers as a complete bipartite graph [1-5]. However, partial overlaps can indeed be an interesting research topic, *e.g.*, for mobile networks.
>
> > The experimentation doesn’t include comparison with classical methods such as LSQ. It would be interesting to see how a distributed, blind, greedy LSQ compares to the distributed MARL method proposed here, especially since the computation costs are so vastly different.
> >
>
> ***Response:*** We respectfully refer the reviewer to our results in Table 1 (L259) and 3 (L293) in the revised main paper, and our results in Sec. E in the supplementary material, where we **compared our proposed framework with three classical methods** — *i.e.*, WCMP, **LSQ**, and SED — in the real-world testbed. We will improve the clarity of the paper and make our comparison with classical methods more explicit in our revision.
>
> ---
>
> Given the clarifications, we kindly ask the reviewer to re-evaluate our paper and consider recommending it for acceptance. We hope to have answered your questions sufficiently and helped you see the merits of this paper in a new light. We look forward to your feedback.
>
> ---
>
> Reference:
>
> [1] Eisenbud, Daniel E., et al. "Maglev: A fast and reliable software network load balancer." *13th USENIX NSDI 16*. 2016.
>
> [2] Barbette, Tom, et al. "A High-Speed Load-Balancer Design with Guaranteed Per-Connection-Consistency." *17th USENIX NSDI20*. 2020.
>
> [3] Olteanu, Vladimir, et al. "Stateless datacenter load-balancing with beamer." *15th USENIX Symposium on Networked Systems Design and Implementation (NSDI 18)*. 2018.
>
> [4] Gandhi, Rohan, et al. "Duet: Cloud scale load balancing with hardware and software." *ACM SIGCOMM Computer Communication Review*  44.4 (2014): 27-38.
>
> [5] Desmouceaux, Yoann, et al. "6lb: Scalable and application-aware load balancing with segment routing." *IEEE/ACM Transactions on Networking* 26.2 (2018): 819-834.

---

> ### Author Response · Authors · 2022-08-08
> **Reminder: updated feedback**
>
> Dear Reviewer cVpL,
>
> Thanks again for your review. We hope our rebuttal response has addressed your questions/concerns. We have also made appropriate modifications in the revised pdf to highlight the fact that we did compare with classical load balancing methods.
>
> We hope you would take a chance to reconsider your rating in light of the points raised in support of the merits of our work.
>
> We are more than happy to discuss if you have any further concerns. Please kindly let us know your feedback.
>
> Thank you for your time!
>
> Paper10297 Authors

---

### Official Review · Reviewer_Y8vv · 2022-07-21

**Rating:** 6
**Confidence:** 5
**Soundness:** 2 fair
**Presentation:** 2 fair
**Contribution:** 1 poor

**Summary:**

This paper proposes MPG-based MARL solution for the load-balancing problem. Applying RL directly for load-balancing is not favorable as the load balancers (i.e., multiple-agents) need to synchronize observations as well as the action space grows with number of agents (requiring re-training etc.)

**Questions:**

Please clarify the questions and limitations raised in the weakness section above.

**Limitations:**

* Limited evaluation
* Strong assumptions for a practical system
* No comparison with other ML-based approaches.

**Strengths And Weaknesses:**

**Strengths**
1. Does not require re-training with increasing number of multiple agents as the proposed approach decomposes
the joint state and action space.
2. Does not require synchronization between the load balancers


**Weakness**
1. Poor evaluation
  * No scaling experiments (i.e., increasing number of LBs or servers). Only 2 LBs in evaluation. If the paper had scaled the experiments more they would find that their approach may not be practical for real DCs (discussed later).
  * No real traffic/workload. The supposedly real benchmark is a mocked up small testbed that does not mimic real distribution of traffic or scale.
  * Experimentally weak: no variation in traffic and limited variation of IO/CPU
  * What about QoS (99th percentile behavior, an important metric to evaluate for LBs)?
2. Invalid and inconsistent assumptions wrt problem statement.
  * In the intro, the paper claims that "existing algorithms are not adaptive to due to dynamic environments" yet the assumption made in this paper is that each server is capable of processing a certain amount of workload v_j, which is a number only dependent on server capabilities and not on the request (or traffic type itself). For example, a GET request of type X can take 2s whereas a another GET request of type Y can take 20s. The paper mentions collided elephants and yet does not provide any experiments that the proposed technique can handle such situations. In other words, v_j should be stochastic and not fixed on just the server but also on the request characristics.
  * previous assumption invalidates most of the derivation presented later in the paper.

  * Another assumption is that active probing is impractical. However, it is okay for LBs to communicate with severs to observe the server state, why? There is no citation or experiment showing that indeed that is a reasonable assumption. All of the work is based on this key assumption.

3. Limited insights:
  * why Markov Potential Games for handling the stated problem? Why not use mean-field theorem to approximate the behavior of all the other multi-agents using mean or median behavior. Overall the paper reads as an application of MPG rather than "there is a nice solution" to the load-balancing problem. Insights and analysis of different approaches are missing.
 * No evaluation of the overhead of the RL vs MARL solution in terms of performance as well as overhead (to justify MARL is needed over RL). There are several solutions proposed such as RLB-SAC (Neurips 2021) which reports similar high performance, and Park: An Open Platform for Learning-Augmented Computer Systems.
 * What happens if RL makes bad decisions (safety of RL: Towards safe online reinforcement learning in computer systems, NeurIPS 2021).

4. Writing needs significant improvement esp. introduction and related work section.
     * citations on key assumptions/claims or experiments to make those statements
    * Abbreviations introduced without describing what it stands for?: e.g. NE for MPG

---

> ### Author Response · Authors · 2022-08-02
> **First response to Reviewer Y8vv (1/2)**
>
> Dear Reviewer Y8vv:
>
> Thank you for your time to attentively review our paper and we appreciate your thorough and constructive comments.
>
> There seem to be a couple of misunderstandings, regarding:
>
> - notations and assumptions in our paper,
> - scaling experiments,
> - real-world traffic, and
> - the way we gather networking features and system state observations.
>
> We will try to rectify what we can here. We will address the comments — point by point — by grouping them into 3 categories: (1) evaluation, (2) assumption, and (3) insights.
>
> ---
>
> ## 1. Evaluation
>
> > No scaling experiments (i.e., increasing number of LBs or servers). Only 2 LBs in evaluation.
> >
>
> ***Response:*** We do agree with the importance of scaling experiments and we respectfully refer the reviewer to our results with 6 load balancers (LBs) and 20 servers in Table 3 in the main paper (description can be found at L293), as well as the overhead analysis in the supplementary material (Sec. E.2.2). We have added the number of LBs and servers to Table 6 in the supplementary material (as LB System parameter) as well for making it more prominent and explicit.
>
> > No real traffic/workload. The supposedly real benchmark is a mocked-up small testbed that does not mimic the real distribution of traffic or scale.
> >
>
> ***Response:*** We appreciate the reviewer for the prospect of real traffic/workload. We did use real traffic and workload and we have clarified our real-world testbed in detail in our answer to the **Common Concern 2** in our **Meta Response** to all reviewers.
>
> > Experimentally weak: no variation in traffic and limited variation of IO/CPU
> >
>
> ***Response:*** We agree that more sensitivity analysis (*e.g.*, by varying CPU/IO, traffic characteristics) will be interesting. However, to thoroughly evaluate what happens with extensive parameter tuning in terms of testbed configuration requires additional modeling, evaluation, and analysis, which, we estimate, would hinder the clarity of this paper. We have already open-sourced both our simulator and testbed in case more researchers would like to join forces for future work.
>
> We have tried to balance our contribution between theoretical modeling and applied evaluation by way of both simulation and experiments running on real machines, which allow for verifying the performance under realistic constraints — *e.g.*, overlayed optimization techniques in the data plane and the network protocol stack, bursty and workloads of long-tail distribution, heterogeneous and dynamically changing server processing capacities.
>
> We have varied the traffic in the simulation with 3 types of workloads, i.e. 100% CPU intensive application, 75% CPU & 25% IO application, and 50% CPU and 50% IO application.
>
> We have varied the traffic in the experimental testbed on physical servers with real-world traffic on two different scales, under different traffic rates, using trace replay from the different hours of the day.
>
> We have also varied server configuration over time (Sec. E in the supplementary material)
>
> However, this comment is well taken, and it is considered for future work.
>
>
> > What about QoS (99th percentile behavior, an important metric to evaluate for LBs)?
> >
>
> ***Response:*** Additional evaluation on a real-world testbed is conducted regarding the 99th percentile of task completion time (QoS). These results are included in the revised supplementary material (Table 9, 10, 11), where our proposed MARL-based load balancing method achieves superior performance than SOTA load balancing methods. Please don't hesitate to check.
>
> > No comparison with other ML-based approaches: No evaluation of the overhead of the RL vs MARL solution in terms of performance as well as overhead (to justify MARL is needed over RL). There are several solutions proposed such as RLB-SAC (Neurips 2021) which reports similar high performance, and Park: An Open Platform for Learning-Augmented Computer Systems.
> >
>
> ***Response:*** We thank the reviewer for providing pointers to these interesting related works. We have added additional experiments to compare our proposed method against the latest solution RLB-SAC (NeurIPS 2021), whose results are also presented in the 99th percentile QoS comparison above. The performance of RLB-SAC is not as good as the proposed distributed method in both moderate-scale and large-scale settings. Our proposed MARL load balancing methods achieve superior results because of (i) a well-designed MARL framework with a carefully selected potential function, and (ii) the use of the recurrent neural network to handle load balancing problem as a sequential problem.
>
> (to be continued)

---

> > ### Author Response · Authors · 2022-08-02
> > **First response to Reviewer Y8vv (2/2)**
> >
> > (cont'd)
> >
> > ## 2. Assumption
> >
> > > the assumption made in this paper is that each server is capable of processing a certain amount of workload v_j, which is a number only dependent on server capabilities and not on the request (or traffic type itself). […] In other words, v_j should be stochastic and not fixed on just the server but also on the request characristics.
> > >
> >
> > ***Response:*** We agree with the reviewer that it is important to take into account the variance of network requests.  However, **there seems to be a misunderstanding about the notations in the paper**. We respectfully refer the reviewer to L102, where we defined $w_i(t) \in W$ (instead of $v_j$) as the notation of workloads (*e.g.*, it can be instantiated as 2s or 20s request), which has **no dependency on server capabilities but on the workload distribution** $W$ itself. **$v_j$, on the other hand, denotes the max processing speed of each server (defined at L108)**, which is dependent on server capabilities. Given $\alpha_{ij}(t)$ as the probability mass of assigning tasks from load balancer $i$ to server $j$, we then represent the residual workload on server $j$ at each timestamp as $\max\{0, \sum_{i=1}^M w_i(t)\alpha_{ij}(t) - v_j\}$ (described at L110). The actual processing times for different requests are indeed different in our system.
> >
> > > previous assumption invalidates most of the derivation presented later in the paper.
> > >
> >
> > ***Response:*** We hope that we have addressed this concern by our answer to the question above.
> >
> > > Another assumption is that active probing is impractical. However, it is okay for LBs to communicate with severs to observe the server state, why? There is no citation or experiment showing that indeed that is a reasonable assumption. All of the work is based on this key assumption.
> > >
> >
> > ***Response:*** There seems to be **a misunderstanding about the way load balancers (LBs) observe server states**. We do agree that autonomously managing network services by active probing servers is impractical in data center networks. We haste to point out that our proposed load balancing mechanism observes the server states by passively processing networking packet headers, instead of actively probing servers.
> >
> > We did not stress this feature collection mechanism — which is one of our key contributions that made possible the deployment of the MARL algorithm in real-world experimental environments — in the main paper because we considered that it might hinder the clarity of this paper. But based on your constructive comments, we believe that appropriate modifications need to be made in the main paper to clarify the fact that our proposed load balancing mechanism collect server state observations passively and incurs no communication overhead. We have put detailed description about our feature collection mechanism in Section C.2.4 in the revised supplementary material.
> >
> > We respectfully refer the reviewer to our answer to **Common Concern 2** in our **Meta Response** to all reviewers above.
> >
> > ---
> >
> > ## 3. Insights
> >
> > > why Markov Potential Games for handling the stated problem? Why not use mean-field theorem to approximate the behavior of all the other multi-agents using mean or median behavior.
> > >
> >
> > ***Response:***
> >
> > This is a valuable remark. We respectfully refer the reviewer to our answer to **Common Concern 3** in our **Meta Response** to all reviewers.
> >
> > > What happens if RL makes bad decisions (safety of RL: Towards safe online reinforcement learning in computer systems, NeurIPS 2021).
> > >
> >
> > ***Response:*** Thank you for this insightful remark and we will clarify this point in our revision. The proposed framework makes load balancing decisions and assigns server j based on the form $j = \arg \min_{k\in[N]}\frac{q_{ik}(t)+1}{a_{ik}(t)}$, where $q_{ik}$ is the observed queue length on server $k$ and $a_{ik}$ is the agent’s action. By design, $a_{ik}$ is within a positive action range. The worst case RL decisions will potentially overload a subset of servers while starving another subset of servers, by assigning bad weights ($a_{ik}$) to servers. This can also happen to classic load balancing algorithms if server weights are misconfigured (see Fig. 2b in the main paper), leading to increased task completion time. We appreciate that the reviewer mentions this problem, but the safety of RL itself is an unsolved research topic, and our work is on a specific application. Therefore, we would put this as our future work instead part of the current paper.
> >
> > ---
> >
> > Given the clarifications and additional experimental results detailed above, we kindly ask the reviewer to re-evaluate our paper and consider recommending it for acceptance.
> >
> > We thank you in advance for your reply. We hope our responses and edits help convince you of the merits of this paper. Please do not hesitate in contacting us during the rolling discussion and let us know if there is anything else that we can revise.

---

> > ### Comment · Reviewer_Y8vv · 2022-08-08
> > **Response**
> >
> > Scaling experiments
> > ---
> > Thank you for pointing out that there were 6 LBs in table 3. However, by a scaling experiment, one really is trying to indentify at what point does the algorithm break (or does not perform better). In table 3, for example one could scale the LBs from 1 to 12 (or 6 if the resources is limited) and #requests from some nominal value of 100 to 100K. Authors are trying to show that they are better than other algorithms (which is okay and required) but they also need to show that for some scale, configuration etc, the algorithm/methodology does break. It is a common practice for a system's work. Given that this paper falls under the category of ML4systems (no innovation on ML side), I would expect this analysis irrespective of what was done before by previous authors in the previously published articles.
> >
> > Real traffic
> > ---
> > I do not see any details wrt real workload in common concerns 2. I do not see why the traffic used is a real world traffic. I would ask authors to drop the idea of calling it a real workload unless they can cite some paper that says that the arrival rate of queries is X based on real user traffic analysis.
> >
> > realism
> > ---
> > e.g., overlayed optimization techniques in the data plane and the network protocol stack, bursty and workloads of long-tail distribution, heterogeneous and dynamically changing server processing capacities.
> >
> > I dont see how these experiements were done and how methodological these experiments were?
> >
> >
> > What about QoS (99th percentile behavior, an important metric to evaluate for LBs)?
> > ----
> > Thank you for these additional experiments.

---

> > > ### Author Response · Authors · 2022-08-08
> > > **Response to Reviewer**
> > >
> > > Dear Reviewer Y8vv,
> > >
> > > Thank you so much for your reply. We are glad to see that the reviewer is happy with our additional experiments regarding QoS. We also would like to point out that we included the SOTA RL-based load balancing method from NeurIPS 2021 (RLB-SAC), whose results are shown in the revised pdf. We hope that the reviewer could merit these contributions as well.
> > >
> > > ## Re: Scaling Experiments
> > >
> > > Thank you for pointing out the experiments that you would like us to conduct. As mentioned in our answer to the **Common Concern 1**, the sets of experiments we conducted have the configurations that correspond to real-world setups, where servers are heavily loaded. We have shown in the current version that scaling up the number of load balancers from 2 to 6, our proposed method still shows superior performance than SOTA load balancing methods. We could understand that the reviewer would like to see how the proposed method behave under stress tests. However, given the remaining time is limited for us to configure the testbed, run additional experiments, and analyze as well as plot the results, we could only try our best and investigate how the performance could potentially drop when facing increasing traffic rates with fixed testbed scale (6 load balancers + 20 servers). We hope that this set of experiments can address your concern regarding scaling experiments. If more time is allowed, we would be happy to conduct more extensive experiments to analyze the system sensitivity. We will post new experimental results as soon as possible and we hope that you would take a chance to reconsider your rating.
> > >
> > > Though, we respectfully argue that this paper do have contributions on the ML side, including the theoretical formulation of network load balancing problem and the design of potential functions. Specifically, the choice of fairness metric as potential function is not straightforward, only the Variance Based Fairness (VBF) can satisfy the Markov potential game assumption and serve as the individual reward function for each LB, with proofs in paper. Moreover, we show that although single-agent RL and vanilla cooperative MARL algorithm like QMIX can be applied for this problem, they do not perform as well as the decentralized learning method we proposed. The theoretical insights also contribute a lot in providing good performance.
> > >
> > > ## Re: Real Traffic
> > >
> > > We are sorry that we forgot to point you to our answer to the **Common Concern 1** as well -- where we cited the paper [1] that says the traffic that we used is from a real-world traffic analysis [2] -- so that our rebuttal on the point of "real-world traffic and testbed configuration" will be complete.
> > >
> > > ## Realism
> > >
> > > By overlayed optimization techniques in the data plane, we meant _e.g._ the kernel bypassing technology as in DPDK [3], loop-unrolling and pre-fetching techniques used in VPP [4]. These techniques are typically used for optimizing high performance network data planes, and they are also employed in our paper when implementing our MARL-based load balancing method. Bursty traffic and long-tail distributed workloads are reflected in our experiments by way of the real-world traffic that we used as in [1, 2]. Heterogeneous architecture exists in cloud computing [5], which motivated us to use 2 groups of servers with different processing capacities (2-CPU vs. 4-CPU). Dynamically changing server processing capacities are evaluated experimentally in Sec. E.2.3 in the supplementary material.
> > >
> > > In the paper, we implemented both an event-based simulator and a physical-machine-deployable testbed. In the simulator, these factors are hard to be captured. Therefore, we used the simulator only for evaluating the theoretical distance between our methods and the theoretical optimal solution. Then we used the experimental testbed deployed on physical machines with real traffic to test the performance of our proposed algorithm under the overlayed constraints from different layers in the network stack.
> > >
> > > ---
> > >
> > > Reference:
> > >
> > > [1] Desmouceaux, Yoann, et al. "6lb: Scalable and application-aware load balancing with segment routing." IEEE/ACM Transactions on Networking 26.2 (2018): 819-834.
> > >
> > > [2] Urdaneta, Guido, Guillaume Pierre, and Maarten Van Steen. "Wikipedia workload analysis for decentralized hosting." Computer Networks 53.11 (2009): 1830-1845.
> > >
> > > [3] https://www.dpdk.org
> > >
> > > [4] https://wiki.fd.io/view/VPP
> > >
> > > [5] Kumar, Adithya, et al. "The fast and the frugal: Tail latency aware provisioning for coping with load variations." Proceedings of The Web Conference 2020. 2020.」

---

> > > > ### Comment · Reviewer_Y8vv · 2022-08-09
> > > > **Response**
> > > >
> > > > Thank you for your response. I will update my score based on this discussion.
> > > >
> > > > I will also suggest authors to write the details of wikipedia replay technique in supplementary materials (or refer the readers to the exact section of the paper if it is exactly the same as [1]). Based on my reading of [1], section 6.c, the methodology used in that paper makes sense.

---

> > > > > ### Author Response · Authors · 2022-08-09
> > > > > **Thanks for for raising the score & Additional scaling experimental results**
> > > > >
> > > > > Dear Reviewer Y8vv:
> > > > >
> > > > > Thank you so much for raising the score from 3 to 6. We appreciate this valuable discussion with you, which has definitely made our paper stronger.
> > > > >
> > > > > ## Wikipedia Replay
> > > > >
> > > > > In terms of the Wikipedia replay technique, we managed to use exactly the same setup as in [1], and we have documented the implementation details in Sec C.2.3 in the supplementary materials already.
> > > > >
> > > > > ## Scaling Experiment
> > > > >
> > > > > As promised, though given limited time, we have conducted additional experiments using 6 load balancers + 20 servers testbed with different traffic rates from low to high. Under low traffic rates, when servers are all under utilized, the advantage of our proposed method is not obvious because all resources are very much over-provisioned. With the increase of traffic rates (till servers are 100% saturated), our methods outperforms the best classical methods. Please check the 4 tables below, comparing both the 99th percentile and average task completion time for both Wiki pages and static pages.
> > > > >
> > > > > ### Wiki Pages - QoS - 99th percentile task completion time (s)
> > > > > | Traffic Rate (queries/s) |  731.534 | 1097.3 | 1463.067 | 1828.834 | 2194.601 | 2377.484 | 2560.368 | 2743.251 | 2926.135 |
> > > > > |---:|---:|---:|---:|---:|---:|---:|---:|---:|---:|
> > > > > | LSQ  | 0.175 +/- 0.015 | 0.212 +/- 0.025 | 0.249 +/- 0.043 | 0.342 +/- 0.121  | 0.827 +/- 0.572 | 2.103 +/- 0.654 | 10.662 +/- 2.557 | 17.656 +/- 0.714 | 17.999 +/- 0.253 |
> > > > > | SED  | 0.201 +/- 0.022 | 0.261 +/- 0.079 | 0.322 +/- 0.099 | 0.360 +/- 0.088 | 0.618 +/- 0.268 | 2.175 +/- 1.328 | 11.444 +/- 3.861 | 22.086 +/- 4.892 | 22.727 +/- 5.632 |
> > > > > | Distr-LB [VBF] | **0.160 +/- 0.010** | **0.205 +/- 0.036** | **0.248 +/- 0.086** | **0.284 +/- 0.113**   | 0.567 +/- 0.306 | **1.276 +/- 0.647** | 7.005 +/- 1.147  | 10.560 +/- 1.042 | 15.745 +/- 0.254 |
> > > > > | Distr-LB [VBF+logVBF] | 0.161 +/- 0.008 | 0.216 +/- 0.052 | 0.249 +/- 0.068 | 0.348 +/- 0.122    | **0.439 +/- 0.121** | 1.533 +/- 0.670 | **4.427 +/- 0.443**  | **9.391 +/- 0.329** | **15.347 +/- 0.572** |
> > > > >
> > > > > ### Static Pages - QoS - 99th percentile task completion time (s)
> > > > > | Traffic Rate (queries/s) |  731.534 | 1097.3 | 1463.067 | 1828.834 | 2194.601 | 2377.484 | 2560.368 | 2743.251 | 2926.135 |
> > > > > |---:|---:|---:|---:|---:|---:|---:|---:|---:|---:|
> > > > > | LSQ  | 0.014 +/- 0.001 | 0.015 +/- 0.000 | 0.015 +/- 0.000 | 0.018 +/- 0.003 | 0.217 +/- 0.305 | 0.856 +/- 0.554 | 11.066 +/- 3.095 | 16.874 +/- 0.391 | 17.155 +/- 0.217 |
> > > > > | SED  | 0.014 +/- 0.000 | 0.015 +/- 0.000 | 0.016 +/- 0.001 | 0.018 +/- 0.001 | 0.071 +/- 0.066 | 1.252 +/- 1.489 | 11.272 +/- 3.975 | 21.941 +/- 5.970 | 20.708 +/- 5.423 |
> > > > > | Distr-LB [VBF] | 0.014 +/- 0.000 | 0.015 +/- 0.000 | 0.016 +/- 0.001 | **0.017 +/- 0.000** | **0.041 +/- 0.025** | **0.338 +/- 0.364** | 6.670 +/- 1.152  | 9.743 +/- 0.863  | 15.506 +/- 0.056 |
> > > > > | Distr-LB [VBF+logVBF] | 0.014 +/- 0.000 | 0.015 +/- 0.001 | 0.016 +/- 0.000 | 0.018 +/- 0.002 | 0.072 +/- 0.087 | 0.465 +/- 0.403 | **3.970 +/- 0.545**  | **8.782 +/- 0.187**  | **15.095 +/- 0.497** |
> > > > >
> > > > >
> > > > > ### Wiki Pages - QoS - avg. task completion time (s)
> > > > > | Traffic Rate (queries/s) |  731.534 | 1097.3 | 1463.067 | 1828.834 | 2194.601 | 2377.484 | 2560.368 | 2743.251 | 2926.135 |
> > > > > |---:|---:|---:|---:|---:|---:|---:|---:|---:|---:|
> > > > > | LSQ  | 0.048 +/- 0.002 | 0.055 +/- 0.003 | 0.059 +/- 0.003 | 0.069 +/- 0.008 | 0.131 +/- 0.070 | 0.643 +/- 0.325 | 1.910 +/- 0.269 | 2.873 +/- 0.215 | 3.545 +/- 0.146 |
> > > > > | SED  | 0.054 +/- 0.001 | 0.061 +/- 0.004 | 0.068 +/- 0.004 | 0.080 +/- 0.004 | 0.117 +/- 0.025 | 0.660 +/- 0.396 | 1.718 +/- 0.366 | 2.767 +/- 0.207 | 3.482 +/- 0.189 |
> > > > > | Distr-LB [VBF] | 0.047 +/- 0.001 | 0.054 +/- 0.003 | 0.059 +/- 0.005 | **0.066 +/- 0.007** | 0.105 +/- 0.035 | **0.266 +/- 0.139** | 1.465 +/- 0.115 | 2.047 +/- 0.145 | 2.704 +/- 0.108 |
> > > > > | Distr-LB [VBF+logVBF] | 0.047 +/- 0.001 | 0.054 +/- 0.004 | 0.059 +/- 0.004 | 0.069 +/- 0.008 | **0.084 +/- 0.009** | 0.413 +/- 0.249 | **1.183 +/- 0.063** | **1.838 +/- 0.083** | **2.513 +/- 0.105** |
> > > > >
> > > > > ### Static Pages - QoS - avg. task completion time (s)
> > > > > | Traffic Rate (queries/s) |  731.534 | 1097.3 | 1463.067 | 1828.834 | 2194.601 | 2377.484 | 2560.368 | 2743.251 | 2926.135 |
> > > > > |---:|---:|---:|---:|---:|---:|---:|---:|---:|---:|
> > > > > | LSQ  | 0.004 +/- 0.001 | 0.004 +/- 0.000 | 0.003 +/- 0.000 | 0.004 +/- 0.000 | 0.018 +/- 0.023 | 0.252 +/- 0.234 | 1.455 +/- 0.258 | 2.426 +/- 0.207 | 3.080 +/- 0.136 |
> > > > > | SED  | 0.003 +/- 0.000 | 0.004 +/- 0.001 | 0.004 +/- 0.000 | 0.004 +/- 0.000 | 0.006 +/- 0.003 | 0.284 +/- 0.308 | 1.283 +/- 0.374 | 2.322 +/- 0.226 | 3.041 +/- 0.188 |
> > > > > | Distr-LB [VBF] | 0.004 +/- 0.000 | 0.004 +/- 0.000 | 0.004 +/- 0.000 | 0.004 +/- 0.000 | **0.005 +/- 0.001** | **0.055 +/- 0.070** | 1.039 +/- 0.144 | 1.617 +/- 0.135 | 2.277 +/- 0.096 |
> > > > > | Distr-LB [VBF+logVBF] | 0.004 +/- 0.000 | 0.004 +/- 0.000 | 0.004 +/- 0.000 | 0.004 +/- 0.000 | 0.006 +/- 0.004 | 0.116 +/- 0.114 | **0.750 +/- 0.063** | **1.413 +/- 0.083** | **2.076 +/- 0.096** |

---

> ### Author Response · Authors · 2022-08-08
> **Reminder: updated feedback**
>
> Dear Reviewer Y8vv:
>
> We hope you are doing well.
>
> Would you mind please giving us an approximate time window as to when you would be responding to our reply? The discussion period with the authors is coming to an end and we would not like to miss your response due to time difference. We have been standing by.
>
> We spared no effort in out first response to address your concern (_e.g._ by including a new baseline in our evaluations, clarifying several concepts that might have caused your 4 major misunderstandings). Corresponding modifications have also been made in our revised pdf. We hope you would take a chance to reconsider your rating in light of the points raised in support of the merits of our work.
>
> What remaining doubts/questions do you have? We'd be happy to clarify and/or discuss.
>
> Thank you very much in advance for your information.
>
> Sincerely,
> Paper10297 Authors

---

### Official Review · Reviewer_XheU · 2022-07-26

**Rating:** 6
**Confidence:** 4
**Soundness:** 4 excellent
**Presentation:** 4 excellent
**Contribution:** 3 good

**Summary:**

This paper focuses on the network load balancing problem in data centers using multi-agent RL paradigm. The main goal in load balancing problems is to minimize the makespan
The authors prove various properties of the setting with the main result to be that such setting is a Markov Potential Game. They showed this result via properly defining a workload distribution fairness potential function. Moreover, using facts established in Leonardos et al, they design a distributed algorithm to approximate Nash equilibrium policies. The authors provide an extensive experimental section that suggest that the proposed algorithm is effective.

**Questions:**

What was the main technical difficulty that did not allow the authors to provide theoretical guarantees for their proposed algorithm?

**Limitations:**

This work has no negative societal impact as far as the reviewer can forsee.

**Strengths And Weaknesses:**

Pros
The paper is interesting, with both theoretical and applied merits and an interesting modeling of the network load balancing prblems in data centers as a MARL system.
Cons: The result of the proposed framework to be a Markov potential game is not very surprising as load balancing games are known to be potential games (see [Koutsoupias, Papadimitriou 99]).

---

> ### Author Response · Authors · 2022-08-02
> **First response to Reviewer XheU**
>
> Dear Reviewer XheU,
>
> Thank you for taking the time to review our paper and we appreciate your insightful comments.
>
> ---
>
> > The result of the proposed framework to be a Markov potential game is not very surprising as load balancing games are known to be potential games (see [Koutsoupias, Papadimitriou 99]).
> >
>
> > What was the main technical difficulty that did not allow the authors to provide theoretical guarantees for their proposed algorithm?
> >
>
> ***Response:*** We appreciate the pointer to the reference. We respectfully refer the reviewer to our answer to **Common Concern 3** in our **Meta Response** above. The authors did not provide theoretical guarantees because in the proposed problem settings the theoretical guarantees would require strong assumptions, which typically fail to capture the reality. It would be encouraged to try to get rigorous theoretical analysis for the algorithm, but due to function approximation and multiple constraints under real-world concerns, it could be hard to do. And the our focus is not on the theoretical-side analysis of the algorithm (convergence bound, etc). The focus of this work is proposing an empirical algorithm to solve the real problem, with a certain level of theoretical insights.
>
> ---
>
> We thank you in advance for your reply. We hope our responses and edits help convince you of the merits of this paper. Please do not hesitate in contacting us during the rolling discussion and let us know if there is anything else that we can revise.

---

> > ### Comment · Reviewer_XheU · 2022-08-07
> > **Thank you for your response**
> >
> > Dear authors,
> >
> > Thank you for your response. The paper definitely has merits, both theoretically and experimentally. I will continue the discussion with fellow reviewers and AC. I feel the paper is slightly above the acceptance bar, mostly for its modelling/experimental part.

---

> > > ### Author Response · Authors · 2022-08-08
> > > **Thank you for your feedback**
> > >
> > > Dear Reviewer XheU,
> > >
> > > Thank you very much for your motivating feedback.
> > >
> > > Layer-4 load balancers play a significant role in data center networks. We believe that our work has advanced the important application of MARL on this problem from both theoretical and empirical perspectives, with clearly superior performance than SOTA load balancing algorithms -- not only heuristic ones in production (_e.g._ ECMP and WCMP are implemented based on the Maglev paper from Google [1]), and RL-based one that was proposed last year in NeurIPS [2].
> > >
> > > Though some reviewers seem to have misunderstandings on our paper - which we totally understand given the time constraints for reviewing huge amount of papers, we are confident with our contributions in this work and we hope to see its acceptance.
> > >
> > > Thank you again for the pleasant discussion.
> > >
> > > [1] Eisenbud, Daniel E., et al. "Maglev: A fast and reliable software network load balancer." 13th USENIX NSDI 16. 2016.
> > > [2] Yao et al. "Reinforced Workload Distribution Fairness." NeurIPS 2021

---

### Author Response · Authors · 2022-08-02
**Meta Response to All Reviewers**

First of all, we would like to take this chance to thank all our reviewers for the valuable reviews. We have put in a major effort to address these comments, questions, and concerns, which we believe has brought the paper to a better level.

The revised paper is uploaded, where all major modifications in the pdf have been highlighted in blue to facilitate the reading. We will also keep updating both the main paper and the supplementary material in the following days during the discussion period if applicable.

Major paper updates as per the reviewer's requests are listed below:

- Added real-world DC specs in the supplementary material (Sec. C.2.6, Table 5) to justify that our testbed configuration meets the “real-world” requirement
- Added motivation for reduced communication overhead (Sec. E.2.2, Figure 9)
- LB system configuration hyperparameter is updated so that the testbed scale (number of load balancer agents and servers) is clear in the supplementary material (Table 6)
- Added technical details regarding the passive feature collection mechanism and complexity analysis in Section C.2.3 in the supplementary material (Figure 6, Algorithm 3, Table 4).
- Added 99th percentile QoS evaluation in the supplementary material (Table 9, 10)
- Added RL-based load balancing method — RLB-SAC (Yao et al. Reinforced Workload Distribution Fairness, *NeurIPS* 2021) — as a baseline in both moderate- and large-scale real-world testbed evaluations (Table 1, 3, 8, 9, 10)
- Writing issues are fixed (*e.g.*, abbreviations introduced, notation $M$, Alg.2 H for Horizon)

We will provide our point-by-point responses to each of your concerns raised in this first set of reviews. However, we have also summarized **4 common concerns** from multiple reviewers, which we would like to address in this thread beforehand. We have noted the reviewers who raised the corresponding concern, yet we would also like to welcome all reviewers to join the discussion.

---

> ### Author Response · Authors · 2022-08-02
> **Common Concern 4 [to Y8vv, yzYr, WwLw]: Writing and Nits**
>
> We thank the reviewers for catching these nits. They have been all fixed. All abbreviations have been checked so that they are introduced on their first appearances.

---

> ### Author Response · Authors · 2022-08-02
> **Common Concern 3 [to XheU, Y8vv]: Why Markov Potential Game**
>
> We agree with Reviewer **XheU** that it is natural to apply a game theoretical approach to tackle resource allocation problems. We appreciate the pointer to the reference [1], which studies the theoretical bounds for the worst-case Nash equilibrium in a basic link load balancing setup.  But we would like to clarify that the network (Layer-4 server) load balancing problem is different from the scheduling problem studied in [1] or the link load balancing problem studied in [2] (the difference between server load balancing and link load balancing problem is also discussed in [3] in the context of data center networks), and it is a challenging problem.
>
> As Reviewer **Y8vv** pointed out, the assumptions in theoretical studies typically have strong assumptions. We summarize the following challenges of applying Markov potential game to Layer-4 server load balancing problem which is studied in the paper:
>
> - As studied in [1, 2, 4], theoretical bounds are derived based on the assumption that resources (links/servers) share identical capacities, and the system is relatively static. With the rise of elastic and server-less computing, where tenants in a data center can share physical resources (*e.g.,* CPU, disk, memory), servers can have different processing capacities, which may also change over time — because of e.g., updated server configuration (upgrading an AWS EC2 a1.xlarge instance to a1.4xlarge, which increases the server capacity by 4x — product detail can be found [here](https://aws.amazon.com/ec2/instance-types/a1/)) or resource contention (co-located workloads) [5].
> - Real-world networking systems consist of multiple layers of technology and overlayed optimization techniques, *e.g.*, random access (a simplified version is studied in [1]), batch processing, loop-unrolling, congestion control, etc. These can hardly be modeled theoretically, which motivates us to implement our framework in a real-world system on physical machines using the flexible learning-based approach.
> - Unlike the assumption in [1] where each agent is aware of its workload $w_i$ to be distributed, or in [4] where every job is identical, Layer-4 server load balancers are agnostic to the information of jobs which they will distribute [3]. Besides, real-world traffic is bursty (*e.g.*, one of the data centers studied in [6] exhibits a median flow inter-arrival time lower than $250\mu$s) and consists of long-tail distribution [7]. Therefore, we exploit recurrent neural networks to condition the policy on the historical data for modeling the distributions and handle this sequential and stochastic problem.
>
> Mean-field theory (*mentioned by Reviewer **Y8vv***) is an interesting approach to handle MARL problem [8], yet it is designed for circumstances where a large number of similar agents co-exist. However, as discussed above, typical real-world setups have less than 15 load balancer agents (which is far less than 100, 500, or 1000 as in [8]). The mean-field treatment for agents with not a very large number may lead to other problems, especially when asymmetric agents are required. Besides, load balancer agents may be different from each other in terms of their infrastructure (*e.g.*, software, commodity switch, or ASICs [9, 10]) or in terms of their placement in the physical data center (whether load balancers are deployed within the same rack as servers [9]). Though mean-field theory does not fit the network load balancing problem, we believe that this clarification is important and will discuss these points in our revision.
>
> ---
>
> Reference:
>
> [1] Koutsoupias, et al. "Worst-case equilibria." 1999.
>
> [2] Sen, Siddhartha, et al. "Scalable, optimal flow routing in datacenters via local link balancing." *CoNEXT*. 2013.
>
> [3] Zhang, Jiao, et al. "Load balancing in data center networks: A survey." *IEEE Communications Surveys & Tutorials* 2018
>
> [4] Goren, Guy, Shay Vargaftik, and Yoram Moses. "Distributed dispatching in the parallel server model." *arXiv preprint arXiv:2008.00793* (2020).
>
> [5] Guo, Jing, et al. "Who limits the resource efficiency of my datacenter: An analysis of alibaba datacenter traces." IEEE/ACM IWQoS. 2019.
>
> [6] Benson, et al. "Network traffic characteristics of data centers in the wild." *ACM SIGCOMM IMC*. 2010.
>
> [7] Roy, Arjun, et al. "Inside the social network's (datacenter) network." *ACM SIGCOMM*. 2015.
>
> [8] Yang, Yaodong, et al. "Mean field multi-agent reinforcement learning." *ICML*. 2018.
>
> [9] Gandhi, Rohan, et al. "Duet: Cloud scale load balancing with hardware and software." *ACM SIGCOMM*. 2014
>
> [10] Miao, Rui, et al. "Silkroad: Making stateful layer-4 load balancing fast and cheap using switching asics." *SIGCOMM*. 2017.

---

> ### Author Response · Authors · 2022-08-02
> **Common Concern 2 [to Y8vv, yzYr, WwLw]: Communication Overhead (1/2)**
>
> We believe that it will be helpful to better motivate reduced communication overhead in real-world data center networks. Therefore, in addition to our analysis in Sec. E.2.2 in the supplementary material as well as the background information about real-world data center networks described above in **Common Concern 1**, we will address this point by clarifying the impact of communication overhead both qualitatively and quantitatively, with additional experiments discussed as follows.
>
> Communication overhead in data center networks can be discussed in two folds: throughput and latency.
>
> - Throughput: Active signaling (*e.g.*, periodically probing, or sharing messages) is an intrinsic way to observe and measure system states so that informed decisions can be made to improve performance [1-3]. Higher communication frequency gives more relevant and timely observations yet there is a trade-off between communication frequency and additionally consumed bandwidth. Especially, in large distributed systems like data center networks, services are organized by multiple server clusters scattered all over the physical data center network in the era of cloud computing. Thus, management traffic among different nodes can cascade and plunder the bandwidth for data transmission in high-tier links. To demonstrate the trade-off between measurement quality and throughput overhead, we have conducted experiments to evaluate (i) the relevance of collected server utilization information to the actual server utilization information with root mean square error (RMSE) and Spearman’s Correlation in our testbed on physical servers. When a controller VM periodically probes a server cluster via TCP sockets (In the 69-byte control packet emitted by the server, the 24-byte payload consists of the server ID, CPU and memory usage, and the number of busy application threads), as shown in the table below, the visibility over system states (relevance between measurements and ground truth) correlates with the probing frequency (also visualized in Figure 9 in the revised supplementary material).
>
>     |Probing Frequency (/s)|CPU (%) RMSE|#Job RMSE|CPU Spearman's Corr.|#Job Spearman's Corr.|2LB-7server (Kbps)|6LB-20server (Kbps)|
>     |---:|---:|---:|---:|---:|---:|---:|
>     |2.2222|48.3335|2.0675|0.2827|0.4651|2.15|18.40|
>     |2.8571|44.5611|1.8466|0.3964|0.5644|2.76|23.66|
>     |4.0000|39.8433|1.6076|0.5197|0.6642|3.86|33.12|
>     |6.6667|32.6542|1.3109|0.6768|0.7729|6.44|55.20|
>     |20.0000|21.9652|0.9056|0.8525|0.8892|9.32|165.60|
>
>     Additional management traffic within a single service cluster — behind one virtual IP (VIP) — can exceed the 90th percentile of per-destination-rack flow rate (100kbps as depicted in Fig. 8a in [4]) in the Facebook data center in production. As depicted in Fig. 9a in the supplementary material, to synchronize and communicate observations among load balancer agents, a substantial amount of data need to be transferred, which can break full-bisection bandwidth (an important throughput-related performance metric) in data center networks [5].
>
> - Latency: The impact of communication overhead in terms of the increased latency is studied in the supplementary material and depicted in Fig. 9b in Sec. E.2.2. It is measured for per-packet round trip time (RTT) between two directly connected network nodes. While normal RTT is 0.099ms +/- 0.014ms in such a setup, with additional communication overhead, RTT can grow more than 10x. This is not considered as low additional latency (*mentioned by Reviewer **WwLw***), especially not in high-performance networking systems. In elastic and cloud computing contexts and real-world setups, load balancers can be deployed in different racks [6]. There can be multiple hops between two nodes and one connection consists of tens of hundreds of packets, which can lead to cascaded high latency.
>
> (to be continued)
>
> ---
>
> Reference:
>
> [1] S. Zhang et al, “Efficient Communication in Multi-Agent Reinforcement Learning via Variance Based Control”, NeurIPS 2019.
>
> [2] Z. Ding et al, “Learning Individually Inferred Communication for Multi-Agent Cooperation”, NeurIPS 2020.
>
> [3] T. Wang et al, “Learning Nearly Decomposable Value Functions Via Communication Minimization”, ICLR 2020.
>
> [4] Roy, Arjun, et al. "Inside the social network's (datacenter) network." *SIGCOMM*. 2015.
>
> [5] Zhang, Jiao, et al. "Load balancing in data center networks: A survey." *IEEE Communications Surveys & Tutorials* 20.3 (2018): 2324-2352.
>
> [6] Gandhi, Rohan, et al. "Duet: Cloud scale load balancing with hardware and software." *ACM SIGCOMM Computer Communication Review*  44.4 (2014): 27-38.

---

> > ### Author Response · Authors · 2022-08-02
> > **Common Concern 2 [to Y8vv, yzYr, WwLw]: Communication Overhead (2/2)**
> >
> > (cont'd)
> >
> > Based on the additional experiment above, we can also see that delayed measurement and communication can cause degraded system state observation. We haste to point out that our proposed load balancing mechanism observes the networking features by passively processing networking packet headers, instead of actively probing servers (***which is a misunderstanding by Reviewer Y8vv***). Based on the collected networking features, we then infer system states with neural networks. We humbly consider this feature collection mechanism as a minor contribution in this paper and we described this mechanism in Sec. C.2.4 with Fig. 6 in the supplementary material. The technical details including the feature extraction and collection mechanism at per-VIP granularity are as follows. We implemented reservoir sampling (for collecting task duration and task completion time) and multi-buffering (for collecting the number of ongoing tasks) with POSIX shared memory, in a high-performance kernel-bypassing programmable data plane (VPP). Observations related to different VIPs are organized in separate POSIX shared memory (shm) files, to manage applications independently. Within each VIP, observations related to each egress equipment (*e.g.*, link or server) are also organized independently. Updating (adding or removing) services (VIPs) and their associated equipment can be achieved by managing different shm files in a scalable way using this design, incurring no disruption on data planes. More detailed technical details can be found in our revised paper in **Sec. C.2.4 in the supplementary material**.
> >
> > To further demonstrate the performance of the passive feature collection mechanism which incurs absolutely **zero communication overhead**, an additional experiment is conducted to compare the feature collection latency. The latency overhead of the passive feature collection process in our paper using [POSIX shared memory](https://man7.org/linux/man-pages/man7/shm_overview.7.html) is compared with different active probing techniques. The idle communication latency is compared using both KVM and Docker containers between two hosts either deployed on the same machine (local) or on two neighbor machines (remote). To compare with the shortest latency possible of a hardware-based SDN controller directly connected to the agent, a loopback test is conducted using a NetFPGA [7] connected to the machine via both Ethernet and PCIe. Aquarius parses features stored in the local shared memory with a simple Python script without generating control messages. Its median processing latency outperforms typical VM- and container-based VNF probing mechanisms [8-10] by more than 94.18*μ*s (also visualized in Figure 10 in the revised supplementary material).
> >
> > ---
> >
> > Reference:
> >
> > [7] Zilberman, Noa, et al. "NetFPGA SUME: Toward 100 Gbps as research commodity." *IEEE micro* 34.5 (2014): 32-41.
> >
> > [8] [ETSI Network Functions Virtualization (NFV) Architectural Framework](https://www.etsi.org/deliver/etsi_gs/NFV/001_099/002/01.02.01_60/gs_NFV002v010201p.pdf). 2014.
> >
> > [9] [OpenStack Project Portal](https://www.openstack.org/). 2019.
> >
> > [10] [OPNFV. Open Platform for NFV (OPNFV) Project Portal](https://www.opnfv.org/). 2019.

---

> ### Author Response · Authors · 2022-08-02
> **Common Concern 1 [to Y8vv, yzYr]: Real-World Testbed**
>
> To justify the setups (*e.g.*, scale, traffic) of our experiment satisfies the “real-world” requirement, we present a brief survey of real-world DC setup based on a set of SOTA load balancing research papers (also added in Section C.2.6).
>
> A modern data center may comprise thousands of servers and hundreds of load balancers (as *mentioned by Reviewer **yzYr***). However, each independent service is exposed in a modular way at one or several virtual IP (VIP) addresses to receive requests, running over a cluster of servers. Each server in the cluster can be identified by a unique direct IP (DIP). Traffic and queries from the clients destined to a VIP are load-balanced among the DIPs of the service. The development of virtualization, where computers are emulated and/or sharing an isolated portion of the hardware by way of Virtual Machines (VMs), or run as isolated entities (containers) within the same operating system kernel, has accelerated the commoditization of computing resources. Therefore, **the gigantic in-production data center network is typically partitioned into small pods**, where different services (VIPs) are hosted. This partition in data center networks makes services more fault tolerant — e.g., when one server cluster (pod) behind a VIP fails, the service is still available on other pods. This scheme also largely motivates the design of our passive feature collection mechanism with POSIX shared memory partitioned by VIPs and DIPs (see Sec. C.2.4 in the supplementary material). Therefore, although our experiments have 2-6 LBs, we believe they already demonstrated a certain level of scalability in a real-world network system.
>
> In networking studies, especially Layer-4 server load balancing problems, “real-world” experiments are conducted at per-VIP granularity. The testbed configurations from SOTA load balancers are summarized in the table below:
>
> | Related Work | Real-World Testbed Scale | Note |
> | --- | --- | --- |
> | 6LB [1] | 2 load balancers + 28 servers (2 CPU each) | Our paper uses the same network trace as input traffic. |
> | Ananta [2] | 14 load balancers for 12 VIPs | The exact number of servers per VIP and the in-production traffic is not documented in the paper. |
> | Beamer [3] | 2 load balancers + 8 servers as small scale and 4 load balancers + 10 servers as larger scale | Large scale experiments are conducted with 700 active HTTP connections max. |
> | Duet [4] | 3 software load balancers + 3 hardware load balancers + 34 servers | Synthetic traffic is applied so that the server cluster behind VIP processes 60k (identical) packets per second. |
> | SilkRoad [5] | 1 hardware load balancer or 3 software load balancers per VIP hosted on PoP (point of presence) cluster. | Real-world PoP traffic is applied, where one server cluster behind VIP processes on average 309.84 active connections per second. |
> | Cheetah [6] | 2 load balancers + 24 servers | A python generator creates 1500-2500 synthetic requests/s as input traffic. |
>
> It’s not practical to always conduct research experiments on dozens or hundreds of servers. Based on the survey above, we believe that the experiments conducted in this paper have met the same standard as in the SOTA load balancers from top venues in networking domain. Using 2 physical servers with 48 CPUs each, we have made our best effort to find a configuration that allows us to conduct experiments similar to real-world setups. Based on the survey above, we believe that the experiments conducted in this paper have reasonable scale — not only in terms of number of agents (2/6 load balancers) and servers (7/20 servers), but also in terms of traffic rates (more than 2k queries per second per VIP and more than 1150.76 concurrent connections in large scale experiments) — and are representative of real-world circumstances.
>
> We would especially like to draw the attention of Reviewer **Y8vv** respectfully, who **may have misunderstood or overlooked our scaling experiments in the main paper**. We would appreciate it if you could reconsider your rating regarding our evaluations in light of the points in support of the substantial efforts for conducting real-world evaluation in our paper.
>
> ---
>
> Reference:
>
> [1] Desmouceaux, Yoann, et al. "6lb: Scalable and application-aware load balancing with segment routing." *IEEE/ACM Transactions on Networking*
>
> [2] Patel, Parveen, et al. "Ananta: Cloud scale load balancing." *ACM SIGCOMM Computer Communication Review*
>
> [3] Olteanu, Vladimir, et al. "Stateless datacenter load-balancing with beamer." *15th USENIX NSDI 18*. 2018.
>
> [4] Gandhi, Rohan, et al. "Duet: Cloud scale load balancing with hardware and software." *ACM SIGCOMM*
>
> [5] Miao, Rui, et al. "Silkroad: Making stateful layer-4 load balancing fast and cheap using switching asics." *SIGCOMM*. 2017.
>
> [6] Barbette, Tom, et al. "A High-Speed Load-Balancer Design with Guaranteed Per-Connection-Consistency." *17th USENIX NSDI20*. 2020.

---

### Meta-Review · Area_Chair_TQCC · 2022-08-22

**Recommendation:** Accept
**Confidence:** Less certain

**Metareview:**

The paper received an uniformly positive evaluation, although all the scores are in the "borderline / weak accept" range. The authors included a long and comprehensive rebuttal and actively participated in the discussion, which made some of the reviewers updating their scores.

I recommend the paper to be accepted, but I understand the decision could be reverted when comparing the paper with the other candidates.

**Award:**

No

---

### Decision · Program_Chairs · 2022-09-14

Accept